# The Small GTPases in Fungal Signaling Conservation and Function

**DOI:** 10.3390/cells10051039

**Published:** 2021-04-28

**Authors:** Mitzuko Dautt-Castro, Montserrat Rosendo-Vargas, Sergio Casas-Flores

**Affiliations:** Laboratorio de Genómica Funcional y Comparativa, División de Biología Molecular, IPICYT, San Luis Potosí 78216, Mexico; maria.rosendo@ipicyt.edu.mx

**Keywords:** small GTPases, Ras, Rho, Rab, Ran, Arf, Miro, BRG, fungi

## Abstract

Monomeric GTPases, which belong to the Ras superfamily, are small proteins involved in many biological processes. They are fine-tuned regulated by guanine nucleotide exchange factors (GEFs) and GTPase-activating proteins (GAPs). Several families have been identified in organisms from different kingdoms. Overall, the most studied families are Ras, Rho, Rab, Ran, Arf, and Miro. Recently, a new family named Big Ras GTPases was reported. As a general rule, the proteins of all families have five characteristic motifs (G1–G5), and some specific features for each family have been described. Here, we present an exhaustive analysis of these small GTPase families in fungi, using 56 different genomes belonging to different phyla. For this purpose, we used distinct approaches such as phylogenetics and sequences analysis. The main functions described for monomeric GTPases in fungi include morphogenesis, secondary metabolism, vesicle trafficking, and virulence, which are discussed here. Their participation during fungus–plant interactions is reviewed as well.

## 1. Overview

Small Ras GTPases (rat sarcoma guanine triphosphatases) are monomeric G proteins of low molecular weight (21 to 30 kDa) that are present in all eukaryotes and that participate in many biological processes. They act as molecular switches, alternating between an active GTP-bound state and an inactive GDP-bound state [1]. This regulation is carried out through the activity of guanine-nucleotide exchange factors (GEFs) and GTPase activating proteins (GAPs) that catalyze the replacement of GDP by GTP and promote the inactive GDP-bound form, respectively (Figure 1) [2,3,4].

These proteins have a conserved GTP/GDP core and a hydrolysis domain bearing five characteristic and highly conserved motifs: G1 (Walker A/P-loop; GxxxxG K[S/T]) implicated in the binding of β- and γ-phosphate groups of the nucleotide; G2 (Switch I; x[T/S]x) that binds Mg^2+^, essential for GTP hydrolysis; G3 (Walker B/Switch II; DxxG) that interacts with the nucleotide γ-phosphate and Mg^2^; G4 ([N/T]KxD) that binds directly to the nucleotide at K and D sites; and G5 (S/CAK/L/T), involved in guanine base recognition [5,6]. The switch I and II regions are flexible segments that sense the nucleotide state and allow the small GTPases to interact with downstream effectors [7,8].

Members of the Small Ras-GTPases superfamily take part in cell regulation, growth, morphogenesis, cell division, and virulence and are known as master regulators of intracellular bidirectional vesicle transport [1,9]. They are classified according to their amino acid sequence and their biochemical properties.

As shown in Figure 1, the most common families and their main functions in fungi are Ras (rat sarcoma), associated with morphogenesis and virulence of fungi with different lifestyles, such as pathogens, saprobes, entomopathogens, and mutualists; Rho (Ras homologue) also known as Cdc42/Rho-GTPases, involved in the regulation of cell polarity, gene expression, and membrane traffic; Rab (Ras homologue from brain) that participates in the secretion of metabolites and lytic enzymes [10]; Arf (ADP ribosylation factors) participate in membrane/protein trafficking to the plasma membrane [9]; Ran (Ras-like nuclear proteins) play pivotal roles in nucleocytoplasmic and nuclear pore transport, synthesis of RNA, and as a checkpoint of the cell cycle [11]; and Miro (mitochondrial Rho-GTPases), also known as atypical Rho GTPases, serves critical roles in mitochondria morphology, inheritance, and homeostasis [12]. Other less studied families have been reported, such as Era-like (*Escherichia coli* Ras-like protein), involved in ribosomal assembly [13], and Spg1/Tem1 (Septum promoting) that participate in ring constriction, septation, and cell division in *Schizosaccharomyces pombe* [14]. Recently, a new fungal family of Ras GTPases was reported, the Big Ras GTPases (BRG), whose main difference is their size, 3- to 4-times bigger compared with the classical small GTPases as well as with their poorly conserved G4 motif. The only member characterized until now, TBRG-1 of the plant mutualistic fungus *Trichoderma virens*, is involved in conidiation, development, secondary metabolism, mycoparasitism, and biocontrol [15]. Proteins from the different families crosstalk with each other and with members of other signal transduction pathways, which together comprise a very large and complex network [9].

In this review, we present a comprehensive work where we discuss the state of the art regarding the main families of monomeric GTPases in fungi. Furthermore, we present analyses at the genomic, phylogenetic, and sequence levels as well as the function of each protein family.

## 2. Classification and Phylogenetic Conservation of Small Ras GTPases in Fungi

Fungi are eukaryotic organisms that are highly relevant ecologically. They differ from plants due to their heterotrophic nature, and from animals because they possess cell walls. Their closest kingdom is the Animalia, from which they diverged nearly 1.3 billion years ago [16,17]. One of their major features is osmotrophy, a way of feeding and nourishment that involves the displacement of degraded organic compounds by osmosis. Fungi secrete lytic enzymes to breakdown the molecules and substrates so they can be passed through their cell walls for feeding [18]. This mechanism is used by many fungal species to compete among themselves or with other microorganisms. Fungi have evolved several strategies to degrade hard-to-digest substrates, which are also useful for establishing a relationship with plants. Moreover, they secrete a plethora of secondary metabolites [19]. The small Ras GTPases play important roles in several of these biological processes in fungi.

During the last two decades, genomic tools have greatly helped to study the phylogenetic relationships between different species of organisms, as well as to analyze complete gene families. With this purpose, James and coworkers [17] analyzed the current state of the fungal tree of life and reported that 224 orders of fungi are distributed into 12 phyla. The Ascomycota phylum contains the greatest number of species, followed by Basidiomycota, which together with Entorrhizomycota form the subkingdom Dikarya. The sister of this subkingdom is Mucoromycota, which contains mostly plant-associated taxa. Other important phyla are Zoopagomycota, related with parasitic and predatory behaviors, and sometimes saprobic; Blastocladiomycota, which is unique, in having an alteration of haploid and diploid generations and whose members have uniflagellated zoospores; and Chytridiomycota, which is composed of Aphelidiomycota, Cryptomycota/Rozellomycota, and Microsporidia, a group endoparasitic zoosporic fungi placed at the basal branches of the fungal kingdom [17].

In this work, we first focus on analyzing the members of the small Ras GTPase superfamily using genomic data. To find which families of Ras GTPases are present in the fungal kingdom and to search for the relationship between their phylogenetic data and their lifestyles, we first selected 56 genomes of different species distributed in eight different phyla (Figure 2, Appendix A). All protein sequences were obtained from “The Fungal Genomics Resource” (MycoCosm. Available online: https://mycocosm.jgi.doe.gov/mycocosm/home, accessed on 14 March 2021) from the Joint Genome Institute (JGI) [20], using three search criteria: (1) PF00071, the Pfam domain key to Ras family, which displays results for the Ras, Rho, Rab, and Ran families; (2) PF00025, ADP ribosylation factor for the members of the Arf family; and (3) PF08477, the Ras complex, Roc, domain of DAPkinase, to find the members of the Miro family. To find the putative Big Ras GTPases, we performed a BLASTP using the sequence of TBRG-1 with each genome analyzed. As shown in Figure 2, the Ras, Rho, Rab, Ran, Arf, Miro, and BRG families are present in fungi, with the Arf family being the most abundant, whereas the Ran family contains the lowest number of members among all species. These patterns of distribution have also been observed in organisms of other kingdoms, such as *Arabidopsis thaliana*, *Oryza sativa*, *Solanum lycopersicum*, *Zea mays*, *Lotus japonicas*, *Medicago truncatula*, *Phaseolus vulgaris*, and *Glycine max* from plants; and *Homo sapiens* and *Drosophila melanogaster* from animals [21,22,23]. Intriguingly, only few species showed a hit with BRG, Ascomycota being the most represented phylum (magenta labels, Figure 2).

With regard to the selected ascomycetes, *Fusarium oxysporum* has the greatest number of Ras GTPases with 57 members compared with the remaining species, whereas the fission yeasts *Schizosaccharomyces japonicus* yFS275 and *Schizosaccharomyces pombe* have only 23 elements each, being the less represented species. In basidiomycetes, *Laccaria bicolor* showed 63 members of Ras GTPases and *Wallemia mellicola* 30, where the first one bears the higher number of Rho GTPases of the analyzed fungi (Figure 2). In the phylum Chytridiomycota, we found the species with the highest number of Ras GTPases. *Neocallimastix* sp. Gf-Ma3-1 bears the highest number with 92 members, being the species with the major number of Ras and Rab families, whereas *Piromyces* sp. UH3-1 has 77 members, with a huge number of Arf proteins (Figure 2). Intriguingly, both species are animal-associated fungi. In Mucoromycota, we found two other species well represented: *Phycomyces blakesleeanus* NRRL1555 and *Mortierella elongata* AG-77, with 65 and 63 elements each, respectively. In contrast, *Encephalitozoon cuniculi* GB-M1 and *Mtosporidium daphniae* UGP3 that belong to the Microsporidia phylum bear 13 and 16 members, respectively, resulting in their being the species with the smallest number of Ras GTPases. Most of the remaining analyzed species ranged between 30 and 40 components of Ras GTPases superfamily, distributed in six families. These distributions in Ras GTPases members correlates well with the total number of genes in each genome (Figure 2, Appendix A). Kelkar and Ochman point out that the wide variation in fungal genomes’ sizes could be related to their lifestyles [24]. In *Laccaria bicolor,* which establishes a mutualistic association with plant roots, the so-called ectomycorrhiza (ECM), it has been proposed that its large genome size resulted from an expansion of gene families’ sizes. This expansion includes protein kinases and Ras GTPases, which overcame a large number of young duplicates or paralogs after the separation from their sister *Coprinopsis cinerea* (saprobe, Figure 2) [25]. Additionally, during this evolution process, in *L. bicolor* members of Ras GTPases have retained, gained, or lost motifs compared with their ancestors, which resulted in the formation of pseudogenes or proteins with new functions. In this genome expansion, those genes related to their symbiotic lifestyle are present [25]. In contrast, members of microsporidia are intercellular parasites of agricultural and medical importance whose genomes are characterized by their reduction in the number of genes. This genome compaction is due to their reduced intergenic spaces and the shortness of most putative proteins relative to their eukaryote orthologues. Furthermore, they lack mitochondria and peroxysomes [26]. In agreement with this, *E. cuniculi* GB-M1 and *M. daphniae* UGP3 here analyzed barely contain 1996 and 3330 genes in their genomes, respectively, and lack Miro Ras GTPases, the mitochondrial Rho atypical proteins (Figure 2, Appendix A).

To investigate the phylogenetic relationships among the seven families of the small Ras GTPases in fungi, we selected 20 species distributed in the eight phyla (denoted by an asterisk, Figure 2), and a single gene of each family was randomly selected to construct a phylogenetic tree (Figure 3). The members of distinct families were grouped, except for putative BRGs, such as those of *Catenaria anguillulae* PL171 and *Rozella allomycis* CSF55, and *Rhizophagus irregularis* A1, which were grouped with Rho and Ran families, respectively. Three proteins annotated in JGI as Ras were grouped with the Rab family (gray labels, Figure 3). As we expected, the subfamilies of each family were also grouped in distinct clades. As shown in Figure 3, the Rab18, Rab1/YPT1, SEC4, Rab5/YPT51, Rab6/YPT6/Ryh1, Rab11/YPT3, Rab2, and Rab4 all from the Rab family are clustered. For the Arf family, we distinguished Arf1, Arl3, Arf6, Arl2, Arl1, and Arf6 to be clustered, with a few exceptions. Similar results were reported by Schmoll et al. (2016) for an analysis of Ras GTPases families using three species of *Trichoderma* (*T. virens*, *T. atroviride*, and *T. reesei*), where the Ras and Rho families derived from the same branch; likewise, both Ran and Arf families derived from the same branch [1]. However, they reported two main branches for these *Trichoderma* species, whereas we found four. This could be explained by the fact that we used a greater number of species with different evolutionary processes. Overall, the proteins of the same phylum were clustered together, being more evident for Ascomycota and Basidiomycota because more species of these phyla were included.

## 3. Structure of Small Ras GTPases in Fungi

Each family of small Ras GTPases belonging to the Ras superfamily has distinct features at the structural level, which, consequently, has repercussions on their functions. They have diverse ways of regulation as well. To gain further insights on these features, we analyzed proteins of the most common families belonging to the Ras superfamily, by including 20 representative species (depicted by asterisks, Figure 2). These results are described in detail below.

### 3.1. Ras

The Ras family is the founding partner of the small Ras GTPases superfamily, which has been the subject of numerous studies, mainly for its role in human oncogenesis [30]. Ras proteins participate in signaling cascades, functioning as signaling nodes that are activated in response to diverse internal and external stimuli. Once activated, Ras proteins interact with several downstream effectors to regulate the cytoplasmic signaling networks to control gene expression and regulate many cellular processes such as cell proliferation, differentiation, and survival [3].

In this study, the 117 Ras GTPases analyzed range from 169 to 310 amino acids in size. The five G-motifs were found at different extents, and all the residues predicted as invariants were perfectly conserved (Figure 4). Another biochemical feature of some families, such as Ras, Rho, Rab, and Arf, is their post-translational modifications by lipids. In this sense, the CAAX motif is located in the C-terminal, where C is a cysteine (C), A represents any aliphatic amino acid, and X are any amino acid [3]. This sequence is recognized by farnesyltransferases and geranylgeranyltransferases I to catalyze the addition of a farnesyl or a geranylgeranyl isoprenoid group, respectively, to the C residue of the CAAX motif. Then, in the endoplasmic reticulum (ER), the proteolytic catalysis of the AAX residue is carried out with the subsequent methylation of the farnesylated C residue [3,31,32]. These modifications are required for membrane binding and subcellular localization; however, second signals are needed, which are added on the so-called hypervariable region (HVR), located immediately upstream of the CAAX [32]. In some cases, the lysine or arginine-rich sequences, also known as polybasic region (PBR), complete the transit of some Ras proteins to the plasma membrane [33]. Furthermore, in other Ras proteins, the covalent addition of palmitate fatty acid to residues, usually one or two C, comprises this second signal [32,34]. Those proteins containing only the CAAX motif are localized to ER, whereas those that also undergo other modifications may be directed along different routes to the plasma membrane. The specific localizations could allow the activation of Ras effector proteins by the different Ras isoforms in a specific and differential way [32].

Here, several of the Ras GTPases analyzed showed the CAAX motif (83/117) at their C-terminal end (Figure 4). The PBR were found in more proteins (43/117) compared with the C residues in the HVR (14/117) related to palmitoylation. The basic residues of PBR can cover up to fourteen residues before CAAX motif, whereas the C-residues were found immediately or a maximum of two amino acids upstream (Figure 4). These two motifs are not present in the same protein.

In filamentous fungi, the small Ras GTPases have been classified in two subgroups, Ras1-like and Ras2-like, where the first bears a CAAX motif, and the second one has been predicted to be geranylgeranylated and bears an HVR. These differences in motifs point to potential distinct subcellular localizations; hence, they probably have single partners under certain environmental and physiological conditions. These characteristics of Ras1-like and Ras2-like contrast with those of the yeasts *Saccharomyces cerevisiae* and *S. pombe*, where Ras1 and Ras2 are highly homologous and bear identical prenylation and palmitoylation motifs in the first one, whereas the second one has only one homologue [36].

### 3.2. Rho

The Rho family has been one of the most studied of Ras GTPases in fungi due to their important roles in morphogenesis and development. The first discovered members in this kingdom were in the yeast *S. cerevisiae* [37], and, up to now, six distinct members have been identified in yeasts: Cdc42 and Rho1–Rho5 [38,39], whereas in filamentous fungi a second subfamily called Rac1 has been reported, which is homologue to Rho5 [1,40]. In *T. reesei*, *T. atroviride,* and *T. virens*, the subfamily members Rho6 and Rho7 have been described. Rho6 has unique features, such as an N-terminal extension region (approximately 130 amino acids) before the G1 motif, and the lysine (K) residue is replaced by a glutamine (G) in said motif. Furthermore, Rho7 appears to be exclusive of *T. virens* [1]. To get more insight into this Rho family classification in fungi, we performed a phylogenetic analysis using the sequences of Rho GTPases of 20 species. We found nine different subfamilies, including Cdc42, Rac, and Rho1–Rho7, distributed in two main branches, one grouping Rho1, Rho2, and Rho4 and the rest in the second one (Appendix A). The most abundant subfamily was Rho1 with 24 members, among 112 proteins analyzed, followed by Cdc42 with 23. Most of the families were perfectly clustered, except for three Cdc42 proteins and two Rho4 (red labels, Appendix A). In agreement with previous reports, we found only one member of Rho7, which corresponded to *T. virens*. Intriguingly, only one Rho6 was found, corresponding also to *T. virens*. Both subfamilies are phylogenetically related to Rho5, where we found only two members. Close to these families, we found three unannotated proteins, where probably *Cladonia grayi* (Rho, JGI ID: 4968) belongs to Rho5 and *F. oxysporum* (Rho, JGI ID: 18381) and *L. bicolor* (Rho, JGI ID: 446900) to Rho7 (vertical red lines, Appendix A). Moreover, the other two unannotated proteins were found clustered independently, whose closest subfamily was Rho2. Furthermore, of seven Rho proteins of *L. bicolor* included in the analysis, five pertain to Rho1 (Appendix A), similar to findings reported by Rajashekar and coworkers [25].

According to our 111 Rho proteins sequences analysis, these range from 151 to 338 amino acids. G1 to G3 motifs are perfectly conserved in most proteins. Interestingly, this family was the only one to have preferentially threonine (T) or leucine (L) residues in G4 instead of K (Figure 4). Moreover, the G5 motif was present in all sequences analyzed. We found the CAAX motif in most of Rho sequences (99/111) as well as in the PBR (76/111), and curiously, only one protein showed the dual cysteine motif (Figure 4).

### 3.3. Rab

Rab proteins regulate budding, transport, and fusion during vesicle transport [41]. Rab proteins contain one or two C residues at the C-terminal as a signal for post-translational prenyl modification. Sometimes these residues are arranged as CAAX motif; however, they could be also arranged as XXCC, XCXC, CCXX, CCXXX, and XCCX. Moreover, all Rab proteins are the substrate for a unique enzyme, the Rab geranyl-geranyl transferase (RGGT), unlike Ras and Rho, where the prenylation motifs determine which prenyl transferase will modify the C-terminal cysteine residues [10]. We analyzed a total of 147 Rab sequences, and the most showed these kinds of C residues at the C-terminal (131/147). Rab analyzed proteins vary in size (174 to 418 amino acids); however, the G-motifs are perfectly conserved among those sequences that showed a hit with them (Figure 4).

Pereira-Leal [42] analyzed the Rab family of 26 species of fungi and reported that the most representative subfamilies are Ypt1, Ypt3, Ypt5, Ypt6, Ypt7, Ypt10, Ypt11, and SEC4, with Ypt5 being the one with the most members. Among the 56 fungal species that we analyzed, the subfamilies annotated in JGI as Rab2, Rab3, Rab4, Rab18, Rab21, Rab26/Rab37, Rab1/Ypt1, Rab5/Ypt51, Rab6/Ypt6/Ryh1, Rab11/Ypt3, and SEC4 are present in fungi. Of these, Rab5/Ypt51 was the most abundant with 135 members, whereas Rab3 and Rab26/Rab37 have only 3 members each (Appendix A). Moreover, Pereira-Leal also reported that the number of Ypt family members ranges from 7 to 12, each of which could be responsible for a specific step of membrane trafficking [10]. However, as shown in Figure 2, we found species with a greater number of Rab family members.

### 3.4. Ran

Ran GTPases are involved mainly in nucleocytoplasmic transport, mitotic spindle assembly, and nuclear envelope assembly [43]. This family of small GTPases has a very particular way of regulation; their functions are dependent on a spatial gradient of the GTP-bound form of Ran, for which the localization of their sites of generation and utilization are essential in this process. The Ran GTPase-activating protein (RanGAP) is localized in the cytoplasm, and the Ran nucleotide exchange factor (RCC1) is in the nucleus. This asymmetric distribution among regulators leads nuclear Ran proteins to be predominantly GTP-bound; Ran in the cytosol is predominantly GDP-bound. This generates a gradient of Ran-GTP across the nuclear envelope that allows control of the directionality of nucleocytoplasmic transport [43,44].

Ran proteins are not subjected to lipid modifications because they lack the C residues at the C-terminal. In addition, they are not anchored in the membrane. However, they do have an acidic tail. It has been shown that deletion of this tail causes that Ran protein to remain primarily GDP-bound at the cytosol, inhibiting the nuclear import of proteins [42]. Ran homologues among organisms of different kingdoms are well conserved, with identity percentages around 80% to each other [1,45]. In agreement with this, among the 59 Ran proteins aligned, we found high conservation in all protein sequences. Additionally, the G-motifs were found in almost all Ran proteins, and all are perfectly conserved. The acidic tail was found in all proteins, characterized mainly by the presence of an aspartic acid (D) residue (Figure 4). This was the most homogenous family according to its size, ranging between 209 and 233 amino acids.

In the 56 species of fungi, almost all have only one member of this family, except for *Mortierella elongata* AG-77 that has three members, and six other species with two members each (Figure 2). Furthermore, their phylogenetic distribution was perfectly clustered according to their phyla (Figure 2).

### 3.5. Arf

Proteins belonging to Arf GTPases are mainly involved in the regulation of coated vesicle budding and their transport. There are three well-characterized types of vesicular carriers involved in the vesicle traffic, which are distinguished according to their coat proteins and their different trafficking routes. The coat complex protein I (COPI) consists of a set of proteins that envelops and shapes the vesicles that move within the Golgi apparatus and from the Golgi back to the ER, called the retrograde transport. The COPII envelops vesicles that are formed in ER and move to Golgi, and clathrin-coated vesicles act in the late secretory pathway as well as in the endocytic pathway [46]. The role of Arf proteins in these processes has been the subject of research for a long time, and specific functions have been assigned for the different subfamilies of Arf. Three subfamilies are identified, including Arf, Arf-like (Arl), and Sar. Arf proteins regulate COPI-dependent retrograde transport and clathrin-dependent budding from trans-Golgi and the plasma membrane. It has been shown that Arf regulate lipid-metabolizing enzymes [47]. Sar proteins are necessary for COPII-dependent transport from ER to Golgi [48,49]. The Arl subfamily has been the less studied; however, it is known that it also functions in membrane trafficking [3]. In 56 fungal genomes, we found a total of 485 Arf homologue proteins (Figure 2), which are distributed in Arf1, Arf6, Arl1, Arl2, Arl3, and Sar subfamilies. From these, Arf1 resulted as the most abundant with 146 members, whereas the rest of the subfamilies were very homogenous in number (53–58), of which, most of the species contain only one member. Interestingly, we figured out that 62 proteins are annotated only as “Arf” (Appendix A), which could pertain to any of the mentioned subfamilies or to others not annotated in JGI, such as Arf5, Arfrp1, or Arl10, which have been reported as *Trichoderma* species [1].

Structurally, the Arf family also has unique features. Most of the Arf proteins are subject to myristoylation, which consists of a co- or post-translational lipid modification in which a myristoyl moiety is attached to a glycine (G) residue localized at the second position from the N-terminal after cleavage of the initial methionine (M). This N-terminal is an amphipathic helix essential for membrane binding and thus for their activity. The association between the myristoyl group and N-terminal amphipathic is inserted into the membrane after GTP binding, leading these proteins to another conformational change and ensuring a very close association with the membrane [49]. These two modifications have been shown mainly in the Arf subfamily, with some exceptions for Arl and Sar. Some Arl such as Arl1 could be N-terminally myristoylated, but most seem to lack this modification despite having the G residue, as has been reported for Arl2 and Arl3 [50]. Moreover, some Arl proteins could be acetylated instead of myristoylated, as reported for Arl3 in *S. cerevisiae* [49]. Sar proteins are not subject to myristoylation; however, the proteins have the N-terminal amphipathic helix, whose hydrophobic residues are necessary for the induction of membrane curvature during vesicle formation [51]. Consistently, our analysis showed that 73 out of 79 Arf sequences have the G residue conserved at the second position, while only 28 out of 58 Arl have it, where the proteins annotated as Arl1 were the most abundant, followed by Arl2 and Arl3 with 18, 8, and 2 sequences, respectively. No member of the Sar subfamily showed the G residue; instead, some hydrophobic amino acids were found at N-terminal helix sequences (Figure 4 and Appendix A). Furthermore, phylogenetically, Arf and Arl in fungi seem to be more related between them. In addition, both subfamilies appear to diverge after Sar (Appendix A). As indicated in Figure 4, Arf proteins also share the characteristic G-motifs of the Ras superfamily. According to protein sizes, the members of this family are homogeneous, ranging from 156 to 272 amino acids.

### 3.6. Miro

Miro is an independent family of the Ras superfamily known as atypical Rho proteins. Miro proteins are localized at mitochondria and regulate the integrity of these organelles [3]; their participation in peroxisomal dynamics regulation has been reported as well [52]. The structure of Miro proteins is unique among all monomeric GTPases, which consists of two GTPase domains flanking two EF-hand Ca^2+^ (EF-H) binding domains, which have a helix-loop-helix structure and a transmembrane domain at the C-terminal to anchor in the outer mitochondrial membrane. In addition, these proteins lack the CAAX motif [12,53]. Two EF-Hs allow Miro to act as a Ca^2+^-dependent switch for mitochondrial movement, thereby permitting the transport when the Ca^2+^ concentrations are basal or low; in contrast, they halt mitochondria when Ca^2+^ levels are high. On the other hand, the GTPase domains are involved in influencing mitochondrial morphology, whereas both EF-Hs and GTPase domains regulate the connections between ER and mitochondria. In relation to G-motifs, it has been reported that the G residue of the G3 motif (DxxG) is not conserved in Miro proteins, compared with the other families [54].

For the Miro fungal family, not much information is available; however, our analysis revealed interesting data. In concordance with previously described data, we found two GTPase domains and two EF-Hs, but with specific features each. In the GTPase domain I, we detected the five characteristic G-motifs but at a different extent, where the G2 was the less present (Figure 5). Except G3, where an alanine (A) residue was found instead of the G residue; all other motifs were well conserved (depicted by an asterisk, Appendix A). EF-hand motifs were conserved according to those reported for other organisms, such as yeasts [53] and humans [12]. Intriguingly, the GTPase II showed less conservation among G3–G5 motifs. In G3, instead of the G residue, we found another aspartic acid (D). In G4, the G amino acid conserved in other families was not present in Miro as in Arf, and the asparagine (N) was not conserved as well. For G5, a valine (V) was found instead of A (represented by asterisks, Appendix A). Most of the analyzed fungal Miro proteins oscillate around 600 amino acids; however, one from *A. niger* (JGI ID: 1187529) and one from *Alternaria alternata* (JGI ID: 118300) are much longer, with 1222 and 1568 amino acids, respectively. Both have extra domains at the C-terminal. In the first case, *A. niger’s* Miro proteins possess a WD40 domain, whereas that of *A. alternata* has a domain related to the transcriptional regulator ICP4 and an activator of mitotic machinery, Cdc14 phosphatase activation domain. In all probablility, all of the changes at the sequence level discussed here influence the activity of Miro proteins; however, more studies are needed.

In overall terms, Miro is a small family compared with other small GTPases, ranging between 0 to 5 members, with *F. oxysporum* f. sp. Lycopersici 4287 being the only one bearing five (Figure 2). Furthermore, from an evolutionary perspective, only Miro of *Dimargaris cristalligena* RSA 468, which belongs to Zoopagomycota, appears to have evolved early compared with the rest of the species (Figure 3).

### 3.7. BRG

Big Ras GTPases are the most recent member of the Ras superfamily reported as of now. The founding member, TBRG-1, was described in the biocontrol fungus *T. virens* and is involved in conidiation and development. TBRG-1 is also a negative regulator of mycoparasitism and secondary metabolism and is important in its biocontrol activity against the fungal plant pathogen *Rhizoctonia solani* in tomato [15].

As depicted in Figure 2, only 10 species showed a hit with TBRG-1 (magenta labels), presenting low identity percentages ranging from 36.2 to 42.8 (Appendix A), as previously reported [15]. In agreement with such report, all proteins are annotated as hypothetical. Five of them also show an annotation related to P-loop containing protein, three with putative aaa ATPase domain, and one with GTPase IMAP family (Appendix A). To further understand this new family, we performed sequences and phylogenetic analysis with these 10 putative BRG. TBRG-1 possesses the G1, G2, G3, and G5 motifs of monomeric GTPases, and the G4 motif is poorly conserved. Indeed, no G4 motif was found among the 10 analyzed sequences using the MEME-suite tool, whereas the other four motifs were perfectly conserved, except for G5, in which the A residue was replaced by tryptophan (W) (Figure 6) [15]. Interestingly, there seems to be a clear evolutionary relationship among the putative members of this family. Those proteins corresponding to *Catenaria anguillulae* PL171 (Can) and to *R. allomycis* CSF55 (Ra), which belong to Blastocladiomycota and Cryptomycota, respectively, two of the early phyla of the fungal tree of life [17,55], seem to be the first members to evolve, followed by the basidiomycete *L. bicolor*. Moreover, these three putative BRG proteins lack G2, G4, and G5 as well as G3 in *R. allomycis* CSF55. The remaining analyzed proteins conserve the G-motifs reported for TBRG-1 and bear other additionally conserved regions among them (Figure 6). In a more interesting way, except for *R. irregularis*, which belongs to Mucoromycota, the remaining proteins pertain to Ascomycota, specifically to Sordariomycetes (bold letter, Figure 6), which is the early-diverging class from this phylum [56], also suggesting an early divergence of the BRG family and possibly a certain specification in their functions related to Sordariomycetes. In agreement, these species share some lifestyles, as most of them can establish a relationship with plants, including *R. irregularis,* and most of them have saprobe lifestyles. Noticeably, besides *T. virens*, the other two species of *Trichoderma* (*T. atroviride* and *T. reesei*) also bear in their genomes a putative BRG, suggesting that this family is conserved among this genus.

## 4. Small Ras GTPases as Molecular Switches with Multiple Functions in fungi

Despite sharing certain features, each small GTPase family has some unique characteristics, which allow them to locate in specific intracellular compartments, conferring them specific functions or acting nearly with a certain frequency in common processes. In agreement with the literature and with our analysis using WoLF PSORT [57] and COMPARTMENTS [58] platforms, the Ras and Rho family proteins are preferentially localized at the ER, cytoplasm, and plasma membrane (Figure 7). Rab GTPases participate in transport and vesicles fusion; consequently, they localize in ER, vesicles, and multivesicular bodies, as well as in early and late endosomes (Figure 7) [59]. Because of their function in nucleocytoplasmic transport, Ran proteins are situated at the nucleus and cytoplasm (Figure 7), while Arf GTPases are founded at ER, Golgi, and cytoplasm owing to their role during the formation and transport of vesicles (Figure 7). By definition, Miro proteins are mitochondrial GTPases, whereas all the BRG proteins showed nuclear localization signals (Figure 7) [57]. Furthermore, Rab, Miro, and BRG GTPases could be localized at the cytoplasm as well but to a lesser extent (Figure 7).

### 4.1. Vesicle Trafficking

Rab and Arf are the main GTPases involved in vesicle formation, trafficking, and secretion. The model yeast *S. cerevisiae* is considered one of the pioneer organisms used in this field. In 1988, the function of Ypt1 in vesicle trafficking was reported [60]. The localization of Sar1 in the ER membrane (Figure 7) as well as its characterization during protein transport and the fact that it is an essential gene in yeast, was described in 1991 [61,62].

In fungi, the secretion process is essential. Saprobic and mycoparasitic fungi secrete a huge number of enzymes needed for breaking down their substrates. Additionally, secretion is necessary in all filamentous fungi for vital biological processes such as the growth of tubular hyphae [62]. Due to this important fact, during the last years, several studies in other fungal species have been published. Furthermore, besides Sar1, it has been shown that other Rab/Arf proteins are essential in fungi.

In *A. niger*, ArfA, which functionally complements *S. cerevisiae*, Arf1/2 is involved in multiple functions of the secretory pathway and is essential since no successful homokaryotic deletion of ArfA strains can be obtained. Using the titratable Tet-on system, it was shown that the absence *arfA* transcript avoids the growth of the fungus, whereas its low expression levels result in an intermediate phenotype, while its overexpression provokes significantly reduced growth and sporulation [62]. The controllable strain, lacking the native *arfA*, is impaired when grown on starch as a sole carbon source, which requires secretion of the amylolytic protein. This strain is also affected when grown in a medium containing calcofluor white and Congo red, which are inhibitors of chitin and glucan assembly. Together, these data indicate that *arfA* is essential and that its regulation must be fine-tuned, which is important in the secretory pathway for transporting cell wall components and biosynthetic enzymes during hyphal growth [62].

Genetic interactions between members of the Rab and Arf families have been reported as well, such as those occurring between Arl1 and Ypt6 in yeasts [63]. Wakade et al. [64] observed that *arl1/arl1* and *ypt6/ypt6* mutants in *Candida albicans* present strongly reduced invasive growth, which was partially rescued by the overexpression of Ypt6 in *arl1/arl1* strain, but not inversely. Furthermore, the *arl1/arl1* mutant has growth defects when exposed to cell wall perturbing agents, whose defect is also rescued by overexpression of Ypt6 for hygromycin B but not for Congo red (cell wall perturbing agents). In agreement with this, both proteins are colocalized during hyphal growth. Intriguingly, the distribution of another Rab protein, Sec4, is altered in the *arl1/arl1*, showing a less tip-clustered compared to wt, whose phenotype is also restored by overexpression of Ypt6 [64].

The relationship between Rab and Arf with other small GTPase proteins during vesicle trafficking has been documented. Cdc42, a protein of the Rho family, plays a key role in the growth of filamentous fungi such as *N. crassa* and *A. nidulans* as well as in *C. albicans*, which switches from budding to filamentous growth in response to external stimuli [65,66,67]. In *C. albicans*, the transient increase of active Cdc42 at the plasma membrane, induced by photo-recruitment, interrupts the membrane traffic and results in de novo secretory vesicle cluster formation immediately after Cdc42 recruitment [64]. Interestingly, some proteins associated with vesicles, including Sec4 and Ypt31 (Rab11 homologue), form part of the cluster. This new vesicle cluster is located at the cell cortex, where the new filament emerges, proving that it is physiologically functional [68].

Together, these data highlight the importance of the small GTPases in vesicle trafficking in fungi.

### 4.2. Morphogenesis

Fungi exist in nature in a diverse variety of shapes appropriate to their different lifestyles. A universal characteristic of fungi is that they are girded by a stable cell wall conformed with proteins, glucans, and chitin [69]. Fungi grow into two main forms: (i) filamentous growth, forming hyphae, and (ii) yeast-like growth [70]. Hyphae are common for most fungi, which are distinguished by continuous tip growth causing the formation of an elongated tube, consisting of a chain of single nucleated cells divided by septa. The yeast cell forms are characterized by discrete cells that divide either by budding or fission, where daughter cells dissociate from the mother cell after division. Interestingly, several mammals or plant pathogenic fungi are able to undergo a dimorphic switch between yeast and hyphal growth shapes; however, the filamentous form is most often related with pathogenicity [71].

Ras-mediated morphogenesis signaling has been widely studied in *S. cerevisiae* and *S. pombe* as well as in the dimorphic fungal human pathogens *C. albicans* and *Cryptococcus neoformans* [72]. In *S. cerevisiae*, Ras1p and Ras2p modulate several processes, such as stress response, pseudohyphal growth, and cell cycle through the cyclic-AMP/protein kinase A (cAMP/PKA) and mitogen-activated protein kinase (MAPK) pathways and Rho-like GTPase networks [73]. Likewise, in *C. albicans*, the cAMP/PKA signal transduction pathway regulates yeast morphology, adhesion, mating, and stress response [74]. Intriguingly, *C. albicans* also grows as pseudohyphae and true hyphae-like molds. This capability of growth transition from yeast-to-hyphae is controlled by Ras1p through cAMP/PKA and MAPK pathways [74]. Contrastingly, in *S. pombe*, Ras1 triggers a MAPK pathway to regulate mating pheromone response and mating together with the Cdc42 pathway via the Cdc42 GEF Scd1 to modulate yeast morphology [36,73]. In *C. neoformans*, Ras1 does not play a critical role in controlling morphogenetic development and virulence through cAMP/PKA pathway; however, this GTPase indirectly interacts with Rho-like GTPase, Cdc42, and Cdc420 as well as with a Rac1 orthologue [75,76].

In contrast with yeast, the signaling mechanisms through Ras GTPases have been poorly studied in filamentous fungi. In *Aspergillus nidulans*, the orthologue of Ras1-like, RasA, was first described, [77]. In this fungus, RasA is essential for growth, negatively regulates conidia germination, is a positive regulator of the initiation of germ tube, and modulates conidiophore development at later growth stages [77,78]. The overexpression of *rasA^G17V^*, the active form of RasA, blocks conidiophore development and low germ tube production phenotypes, which are abolished when levels of *rasA* are diminished [77]. Furthermore, *gapA*, which presumably codes for a GAP for RasA, is potentially a negative regulator of RasA since its absence resembles the phenotype of *rasA^G17V^* overexpressing strains. Concluding, RasA activity must be regulated temporally during *A. nidulans* development.

*Neurospora crassa*, bears *ras-1* and *ras-2* homologous genes on its genome. RAS-1 plays an import role in growth and development, which was unveiled by the dominant mutant allele in the band (*bd*) mutant strain that has been long used as genetic tool for studying circadian rhythms because of its conidial banding pattern in response to blue light [79]. Belden et al. [80] showed that the *bd* mutant is in RAS-1 and bears a point mutation (T79I), having a small increase of GDP/GTP exchange. An increased expression of conidiation-specific genes and altered light-responsive gene levels are present in the *bd* mutant as well. In *N. crassa*, RAS-2 regulates hyphal branching, conidiation, cell wall formation, and growth [81]. Furthermore, RHO-4 GTPase is necessary for regulating both acting and microtubule cytoskeleton, which are important for optimal hyphal tip growth, septum formation, and nuclear distribution [82,83]. Furthermore, GDI (guanine disassociation inhibitors) for RHO-4 regulate localization of RHO-4 to the plasma membrane, septation, growth, and cytoplasmic bleeding [83]. Loss-of-function mutants present few cytoplasmic microtubules and display aberrant nuclear morphology. Microtubules present altered dynamics in *rho-4* lacking strains and in strains bearing GTP-hydrolysis defective or GDP-biased rho-4 alleles. These data indicate that RHO-4 plays an important role in microtubules morphology and dynamics [84].

In *Aspergillus fumigatus*, the most prevalent filamentous fungal pathogen of immunocompromised hosts, *rasA* and *rasB* lacking mutants revealed basically what is known in *A. nidulans* [72]. In *A. fumigatus*, the *rhbA* gene was identified as an upregulated transcript when the fungus was grown in the presence of mammalian cells. Thereafter, Panepinto et al. [85] demonstrated that RhbA regulates growth in medium containing poor nitrogen sources, asexual development in submerged cultures, and virulence in a mouse model of invasive aspergillosis. RasA localization analysis has shown that after post-translational modification it is mainly located to the plasma membrane (Figure 7), and mutation of the highly conserved cysteines necessary for palmitoylation or from the CAAX-box required for prenylation, promotes mislocalization of RasA to endomembranes and to the cytoplasm, respectively. Plasma membrane-located RasA modulates hyphal morphogenesis, cell wall integrity, asexual development, and virulence, whereas endomembrane-localized RasA leads to considerably more hyphal growth than cytoplasmic RasA, indicating that this protein may conserve the capability to dock signal partners to endomembranes [86,87].

*Eremothecium gossypii*, also known as *Ashbya gossypii*, is a filamentous fungus closely related to yeast, first described as cotton pathogen, which also infects other agricultural crops, such as citrus fruits [88]. The Cdc42 Rho GTPase is required for switching from isotropic to filamentous growth during spore germination and formation of hyphal tips where the protein localizes in this fungus [89]. *A. gossypii* has two paralogues, *Ag*Rho1a and *Ag*Rho1b. *Ag*Rho1b potentially regulates cell wall biosynthesis and cellular integrity, whereas *Ag*Rho1a plays a pivotal role in preserving cellular integrity under high hyphal growth speeds [90]. Furthermore, the Rho3 protein works in polarity maintenance [90]. In *A. gossypii*, the *Ag*Rho2 regulates tip-branching, whereas *Ag*Rho5 modulates acting rings at septation sites, mycelial growth, and growth axis [91]. Furthermore, the GEF orthologous to a Rac-GEF from other organisms shows a similar phenotype as *Agrho5* lacking mutant, which indicates that both proteins participate in the same pathway [91]. In this fungus, the Ras-like GTPase Rsr1p/Bud1p is localized at the hyphae tip and plays a role in apical polarization of the actin cytoskeleton, a decisive factor of growth direction (Figure 7). Lack of *rsr1/bud1* provokes a slow growth rate, and hyphal filaments develop aberrant branching sites by mislocalization of the polarisome member AgSpa2p, thereby distorting hyphal morphology. Together these results indicate that Rsr1p/Bud1p is a key player in hyphal growth guidance [92].

*Mucor racemosus,* a zygomycete human pathogen, bears three Ras proteins encoding genes on its genome (*Mras1*, *Mras2*, *Mras3*). *Mras1* messenger is highly expressed during germination and hyphal growth, whereas *Mras3* messenger is highly expressed during sporulation. Regarding the expression of *Mras2*, it has not been detected [93,94]. *M. racemosus* MRas3 regulates germination and growth [94]. Moreover, MRas3 protein but not MRas1 has a crosstalk with the PKA pathway. MRas1 is mainly localized in the cytoplasm and is associated with membrane fractions, whereas MRas3 is mainly located in membranes [94].

In the dimorphic fungal pathogen *Penicillium marneffei,* RasA modulates growth, inhibits aerial hyphae during early growth, and initiates sexual development. Intriguingly, both active and inactive forms of RasA inhibit conidial germination and aberrant hyphal phenotypes under carbon deprivation and produce swollen, misshaped yeast cells at 37 °C [95]. Co-expression of the constitutively RasA and the active version of the Cdc42 orthologous CflA^G14V^ reestablishes germination at the levels of wild-type and complements the abnormal hyphal phenotypes [95].

In the dimorphic human pathogen *Paracoccidioides brasiliensis*, addition of a farnesyltransferase inhibitor, which should mislocalize and inhibit activation of Ras GTPases, enhances the quantity of yeast cells undergoing the yeast-to-hyphae transition during temperature-mediated conversion, indicating a negative role of Ras proteins in dimorphism in *P. brasiliensis* [96]. Haniu et al. [97] showed that yeast cells with low levels of H_2_O_2_ or NO_2_ enhance levels of active Ras post-transcriptionally. Considering that low concentrations of H_2_O_2_ and NO_2_ may function as proliferative signals in fungi, it is tempting to speculate that these molecules may act through the Ras proteins to modulate proliferation by activating small GTPases.

Regarding plant fungal pathogen, few reports exist. In *Colletotrichum trifolii*, the causal agent of anthracnose in *Medicago sativa* (alfalfa), Ct-Ras1 plays important roles in colony morphology, hyphal polarity [98], and accumulation of reactive oxygen species (ROS), which is alleviated by the addition of proline. The antioxidant effect seems to be due to an increase in catalase activity after prolonged culture [99]. Ha et al. [100] reported that Ct-Ras1 positively regulates germination and growth. Additionally, CtCdc42 is partially functionally equivalent to yeast *S. cerevisiae* Cdc42p since it restores the temperature-sensitive phenotype of a yeast Cdc42p mutant. CtCdc42 participates in appressorium differentiation, spore germination, hyphal growth under nutrient deprivation, and accumulation of ROS. This indicates that CtCdc42 is a downstream effector of CtRas [101]. Chen and Dickman [99] showed that Rac antagonizes Ras signaling in *C. trifolii* growth and germination through the MAPK pathway, whereas Ras and Rac act together to produce ROS by a cPLA2 signaling pathway.

Loss of *RAS2* in the maize and barley pathogen *Fusarium graminearum* leads to mate inhibition by impeding perithecia formation. Furthermore, RAS2 participates in the radial outgrowth of colonies, aerial hyphae, and germination. Intriguingly, the absence of *RAS2* does not affect the capability of *F. graminearum* to colonize maize and wheat seedlings [102]. In the filamentous fungus *Magnaporthe grisea*, the causal agent of rice blast, RAS2, participates in appressoria development, whereas RAS1 (a Ras2-like orthologous) regulates conidiation, the only appreciable phenotype. It is noteworthy to point out that loss of the MAPK pathway adaptor protein Mst50 in a RAS2^G23V^ background (the active form), reverts the premature formation of appressoria [103]. Furthermore, the MAPK and cAMP signaling pathways are over-stimulated in a RAS2^G23V^ background, and loss of any integrant of the Mst11-Mst7-Pmk1 signaling pathway inhibits Ras2-mediated appressoria formation [103,104]. In conclusion, Ras1-like orthologue Ras2 potentially works upstream of PKA and MAPK pathways to drive asexual and pathogenic development of *M. grisea*.

*Botrytis cinerea* is a necrotrophic fungus commonly known as gray mold, which affects several plants, including important crops. In this fungus, BcRAS2 is positively involved in hyphal growth, germination, and cell polarity. Notwithstanding such roles of BcRAS2, the fungus is capable of infecting bean leaves, although with a delayed disease occurrence. Furthermore, BcRAS2 participates positively in the concentration of cAMP in the cytoplasm of germlines, and addition of cAMP alleviates the defects in conidia germination and partially reestablishes the reduced growth rate in the Δ*bcras2* strain [105]. In contrast, BcRAS1 plays pivotal roles in fungal virulence and in its tolerance to osmotic and oxidative stresses. Additionally, BcRAS1 regulates positively the expression of genes downstream of the stress regulatory MAPK, BcSAK1, as well as it being hyperphosphorylated under stress conditions [105]. These results indicate that BcRas1 is upstream of the MAPK signaling pathway and that it is the main mediator of Ras-regulated growth and virulence, whereas BcRas2 plays a minor role.

*Setosphaeria turcica* is the main causal agent of northern leaf blight of maize. In this fungus, StRas2 plays a pivotal role in mycelial growth, conidiation, polarity maintenance, appressoria development, appressorial turgor pressure, and virulence. Addition of cAMP to the growing medium does not alleviate defects in hyphal phenotype but partially recovers conidiation and does not restore the appressoria development affectations in a *StRas2* knockdown strain [106]. Together these results may indicate that StRas2 works through the cAMP/PKA signal transduction pathway to control asexual development but regulates hyphal morphogenesis and virulence by a cAMP independent way.

In the entomopathogenic fungus *Beauveria bassiana*, whose biocontrol capability against pests depends essentially on its own virulence and multi-stress tolerance, *Ras1* and *Ras2* are upregulated upon exposure to cell wall disruptive agents, to oxidating compounds, and to the antifungal chemical carbendazim. When subjected to hyperosmotic stress, *Ras2* expression is activated and *Ras1* results are downregulated [107]. In this entomopathogen, Ras1 is involved in growth, hyphal morphology, and asexual development, whereas Ras2 is a main player in hyphal growth and morphogenesis. Lack of *Ras2* also provokes multiple germ tubes at early time-points, implying that *Ras2* may also act to limit polarity, establishing axes early in growth. Furthermore, Ras1 participates in response to oxidative, cell wall, and hyperosmotic stresses and in tolerance to UV light [107]. These phenotypes are associated with deregulation of stress-responsive genes; nevertheless, the stress profiles of each mutant are different, which indicates shared and unique roles for both Ras1 and Ras2 in response to stress. Intriguingly, none of the mutants display abnormalities in tolerance to thermal stress. In conclusion, Ras1 and Ras2 participate antagonistically or differentially in germination, growth, asexual development, multi-stress tolerance, and virulence in this entomopathogen [107].

The filamentous fungus *Trichoderma reesei* is well known as one of the most industrially important microorganisms for its capability to secrete a large quantity of cellulolytic enzymes. This fungus bears on its genome two *Ras* homologues, *TrRas1* and *TrRas2* genes. TrRas1 is involved in colony growth, conidiation, and hyphal polarity. Although the disorder in hyphal growth polarity by the absence of *TrRas1*, *T. reesei* keeps its capability to degrade cellulose. TrRas2 regulates colony growth, conidiation, and hyphal polarity as well, however to a minor extent compared with TrRas1 [108]. Importantly, TrRas2 is a positive regulator of the cellulase cellobiohydrolase’s activities and cellulase protein accumulation. Importantly, the expression of main cellulose-degrading-related genes is regulated by TrRas2 in an Xyr1-dependent way, the master transcriptional regulator of cellulase expression in this fungus [108]. The steady state levels of cAMP are low in Δ*TrRas1* strain but increased in a strain expressing the active TrRas2 version. These data indicate that *TrRas1* is the main regulator of fungal development and could share some roles with TrRas2 on PKA signaling; however, the latter seems to be involved in a single role in modulating cellulase activity in *T. reesei*.

*Schizophyllum commune* is an agarical mushroom-forming fungus. This fungus has been successfully modified genetically and used as a molecular tool for studying cell wall biogenesis, hyphal fusion, and development. This fungus bears two Ras homologues, Ras1 and Ras2; however, due to the order of discovery, Ras1 aligns better with Ras2-like group, while Ras2 aligns with Ras1-like group. In *S. commune*, Ras1 plays an important role in colony growth, apical growth of hyphae, and polarity [109]. Intriguingly, transformants expressing the constitutive inactive version of Ras1 could not be obtained, which indicates that loss of *Ras1* is lethal for this fungus, which results in a distinction compared with other filamentous fungi because Ras2-like is not usually essential in other microorganisms [109]. In *S. commune*, Ras1 positively modulates fruiting bodies formation, which contrasts with previous studies showing that GAP1 positively regulates fruiting bodies formation as well since in its absence this fungus does not fully undergo clamp closure, forms an increased number of fruiting primordia, develops fruit bodies with aberrant gills, and does not produce spores. Analysis of Ras GTPAses signaling and their relationship with PKA reveals that Ras may probably not signal upstream of PKA through cAMP accumulation [109,110]. Despite these results, PKA activity is enhanced significantly in a Δ*gap1*/Δ*gap1* or constitutively active Ras1 transformant, which may indicate that Ras and PKA potentially interact. DNA microarray analysis of gene expression reveals a number of genes coregulated by Gap1, Ras1, and Cdc42, including a putative regulator of mating, transcription, autophagy, and ubiquitination. These data point toward a common signaling pathway for these proteins [109].

*Epichloë festucae* is a mutualistic symbiont (endophyte) of temperate perennial ryegrasses, to which it confers several benefits. In this fungus, NoxR, the regulatory subunit of NADPH oxidases, interacts with RacA, and a mutation of R101E in NoxR abolishes such interaction. Importantly, the mutant allele of *noxR* is uncapable of complementing the absence of *noxR* in planta [111]. Tanaka et al. [112] showed that RacA is necessary for NoxA activation and regulates production of ROS to sustain a symbiotic interaction. *E. festucae* strains lacking *racA* present extensive colonization of its host plant, leading to stunting and premature senescence of seedlings. RacA is a negative regulator of ROS production and aerial hyphae production and positively participates in radial growth. These data indicate that RacA regulates ROS production by NoxA to control hyphal morphogenesis and growth of *E. festucae* in planta.

In the biocontrol fungus *T. virens*, TBRG-1, an atypical Ras GTPase, regulates positively conidiogenesis and conidia development [15]. Furthermore, TBRG-1 participates in the regulation of conidiation-related genes *con-10*, *con-13*, and *stuA*. In *N. crassa*, *con-10* and *con-13* are preferentially expressed during conidia development, whereas StuA is a basic helix-loop-helix protein involved in conidiophore development. This indicates that TBRG-1 is necessary for proper conidia and conidiophore development [15].

### 4.3. Secondary Metabolism

Secondary metabolites (SMs) are bioactive molecules derived from central metabolic pathways and primary metabolite pools [113]. These bioactive compounds are produced by filamentous fungi, plants, bacteria, algae, and animals. Fungi-specific taxa predominantly belonging to the Pezizomycotina Ascomycete class and several Basidiomycete classes are outstanding producers of SMs. SMs play important roles in fungal development as well as in shaping interactions with other organisms.

In cells, endomembrane trafficking machinery, a highly conserved mechanism, is necessary and is shaped by Rab GTPases, docking (tethering) factors, and SNARE (soluble N-ethylmaleimide-sensitive factor attachment protein receptor) proteins and includes an orderly series of events, such as priming, tethering, and fusion [114,115]. This process implies the participation of trafficking vesicles, which occur by membrane budding of the forerunner organelle, and then such vesicles fuse with the destination organelle [115]. Vesicle vacuolar fusion is a conserved transport mechanism for bringing proteins to vacuoles. In yeast, Rab7 the orthologue of Ypt7 is mainly localized in the vacuolar membrane (Figure 7) and is necessary for vesicle docking and vacuole-to-vacuole fusion [116]. In filamentous fungi, secondary metabolism uses this transport machinery as well [117]. The orthologous proteins of Rab7 from *Fusarium graminearum* and *Magnaporthe oryzae*, FgRab7 and MoYpt7, are localized to vacuolar membranes and regulate the fusion of vacuoles and autophagosomes [118,119]. Several authors have shown that the Mon1–Ccz1 complex functions as the GEF for Ypt7 in yeast [120]. In yeast, the absence of either CCZ1 (vacuolar protein trafficking and biogenesis associated) or MON1 elicits vacuole fragmentation [121], a very similar phenotype to the *ypt7* mutant [122]. Furthermore, in *M. oryzae*, the yeast Mon1 orthologue MoMon1 plays pivotal roles in fungal development, pathogenicity, vacuolar assembly, and autophagy [123].

The saprophytic fungus *Aspergillus parasiticus* is mainly found outdoors in areas of rich soil with decaying wood as well as in dry-grain warehouses. This fungus is one of three fungi that produce aflatoxins, a well-known carcinogenic mycotoxin. *A. parasiticus* employs vesicle transport machinery to deliver to the vacuole as well as to synthesize, compartmentalize, and export aflatoxin [117]. The disruption of the *avaA* (formerly vb1) gene that encodes an orthologue of Ypt7 in yeast and AvaA in *A. nidulans*, which is necessary for vacuole biogenesis, results in fragmented vacuole morphology like that observed in the *avaA* mutant in *A. nidulans*. Vb1 contains 205 aa and is 70% identical to Ypt7, 74% identical to mammal Rab7 GTPase, and 94% identical to *A. nidulans* AvaA. Sortin3 blocks the activity of Vps16, a protein of the class C Vps complex in yeast. Addition of Sortin3 to *A. parasiticus* resembles the phenotype of *avaA* mutant [117]. These data indicate that vesicles represent one primary site for the late steps in aflatoxin synthesis, compartmentalization, and export to the cell exterior.

*F. graminearum* is a filamentous fungus that causes the devastating and economically relevant head blight of wheat and related species. The mechanism of vesicular transport is not yet fully known, and considerable advances have been made on Rab GTPases in this fungus. Zheng et al. [118] analyzed all 11 FgRabs by live cell imaging and genetic analysis. They found that FgRab51 and FgRab52 play important roles in endocytosis; FgRab7 is localized to the vacuolar membrane and modulates the fusion of vacuoles and autophagosomes, whereas the results of FgRab8 and FgRab11 are essential for polarized growth and exocytosis. Additionally, both endocytic and exocytic FgRabs are necessary for conidiogenesis, vegetative growth, sexual reproduction, deoxynivalenol metabolism, and pathogenesis in this fungus. Together these data may indicate that Rab GTPases play relevant roles in membrane trafficking-dependent growth and pathogenicity in *F. graminearum* [118].

Furthermore, *F. graminearum* is an extreme threat to human and animal health because the infected cereals are frequently contaminated with deoxynivalenol (DON) and zeralenones (ZEA) [124]. In *F. graminearum*, FgMon1 acts as GEF of FgRab7 through physical interaction. These two proteins play pivotal roles in endocytosis, vacuole fusion, and autophagy; they also control growth, asexual/sexual development, plant infection, and DON production [125]. FgMon1 regulates the DON biosynthesis through the modulation of *TRI5* and *TRI6* genes. Interestingly, FgMon1 is localized in vacuoles of *F. graminenearum* wild-type strain, and its absence provokes vacuoles fragmentation and absence of endocytosis, indicating an important role of FgMon1 in endocytosis and vacuole fusion. Since autophagy is related to membrane trafficking and fusion events and SNARE and Rab GTPases are involved in these processes, the absence of *Fgmon1* shows defects in autophagy under starvation-induced conditions due to the absence of autophagic bodies. In conclusion, FgMon1 could acts as a GEF of FgRab7 by direct interaction in *F. graminearum* [125].

In *Penicillium chrysogenum*, PcvA, the orthologue to AvaA, plays pivotal roles in vesicle–vacuolar fusion. The overexpression of *pcvA* led to halve the vacuole number, pointing to the stimulation of vesicle–vacuolar fusion. Furthermore, the overexpression of *pcvA* leads to diminished conidiation and a significant decrease in production of penicillin, suggesting that vesicular and vacuolar systems play relevant roles in conidiation and secondary metabolism [126]. Interestingly, *P. chrysogenum* shows a different phenotype from that reported for *avaA* overexpression in *A. nidulans* [127], which may suggest differences in the vesicle–vacuolar fusion mechanism for regulating secondary metabolism and conidiation.

*Monascus* spp. is a genus of filamentous fungi used as one of the most important edibles; it can produce many beneficial SMs, such as monacolin K, *Monascus* pigments, and γ-aminobutyric acid among others [128]. In the *Monascus ruber* M7 strain, MRYPT7 protein, the orthologue to YPT7 from yeast, is a positive regulator of radial growth and sexual and asexual development, whereas it participates as a negative modulator in the production of SMs. Transcriptome analysis revealed that MRYPT7 regulates several genes involved in vegetative growth, conidiogenesis, secondary metabolism biosynthesis, and transport in *M. ruber* M7 [129].

Ras is related to the cAMP/PKA signal pathway in yeast; however, the germination program seems to be independent of PKA, although it is partially necessary for the synthesis of SMs, which are involved in *A. nidulans* development. In this fungus, RasA is a negative regulator of *aflR* that codes for a positive regulator of sterigmatocystin biosynthetic genes at both transcriptional and posttranscriptional levels [130]. RasB, which belongs to the Ras2-like group, regulates hyphal branching, conidiophore formation, and radial growth [131].

In *T. virens*, a well-known biocontrol agent the synthesis of secondary metabolites is one of its main traits for controlling phytopathogens. In this fungus, TBRG-1 plays a negative role in antibiotics production [15], which correlates well with the upregulation of secondary metabolism-related genes in the absence of *tbrg-1*, particularly *gliP*, which agrees with an increase in gliotoxin production. *gliP* codes for a non-ribosomal peptide synthase and is part of the gliotoxin biosynthetic gene cluster. These results indicate that TBRG-1 plays a negative role in secondary metabolism [15].

All these data indicate that fungi use conserved cellular protein trafficking mechanisms to synthesize and export secondary metabolites; however, major efforts are necessary to finely dissect this intricated process.

### 4.4. Virulence

The regulation of fungal virulence mediated by small Ras GTPases has been one of the most studied topics, which has allowed an understanding of pathogenic mechanisms.

In *C. albicans*, among their three Arf1–Arf3 and Arl1 and Arl3 proteins, only Arf2 and Arl1 are critical for virulence, whereas Arf2 is required for viability and sensitivity to antifungal drugs [132]. Homozygous deletion mutants of all ARF/ARL genes were successfully generated, except for *ARF2*. For this reason, the tetracycline repression system was used to generate *arf2*Δ/*pTetARF2* mutants (Δ/*pTetARF2*). The repression of *ARF2* leads to defects in filamentous growth and cell wall integrity and virulence, probably by perturbation of the Golgi. Arl1 plays an important role in invasive filamentous growth since its corresponding homozygous mutant shows shorter hyphae and is uncapable of maintaining hyphal growth at a single site. To prove their role in virulence, the mutant strains Δ/*pTetARF2* and *arl1/arl1* were tested in two murine infection models. In the hematogenous disseminated candidiasis (HDC) model, an increased survival percentage was observed in mice injected with both mutant strains after three weeks of infection, while all those mice infected with the wild-type strain died within 11 days. In the oropharyngeal candidiasis (OPC) model, a reduction in virulence was also documented in both cases, observing less oral fungal burden for Δ/*pTetARF2* and *arl1/arl1* mutants, compared with wt [132].

In *F. graminearum,* the lack of RAS2 shows reduced virulence. Interestingly, the phosphorylation levels of the MAPK protein Gpmk1 is considerably low in the Δ*RAS2* mutant background, and the expression levels of FGL1 that encode for a Gpmk1-mediated lipase necessary for virulence is downregulated by the absence of *RAS2* [102]. These data indicate that Ras2 and Gpmk1 act in the same pathway to modulate specific pathogenic traits in *F. graminearum*.

On the other hand, some fungi, such as *Trichoderma* spp., are mycophagous, which gives them competitive advantages against other fungi. In *T. virens*, TBRG-1 participates as a negative regulator of antagonistic and mycoparasitic capabilities against the fungal pathogens *R. solani*, *Sclerotium rolfsii*, and *F. oxysporum*. Furthermore, this Big Ras GTPase negatively regulates mycoparasitism-related genes *sp1* (serine-protease) and *cht1* (chitinase). Together, these data indicate that TBRG-1 plays a pivotal role as a negative regulator of antagonism, mycoparasitism, and the mycoparasitism-related genes in *T. virens* [15].

In *S. cerevisiae*, two Arf-GAPs, Gcs1p, and Glo3p are necessary for retrograde transport, and Glo3p is also required for endocytic recycling of cell surface proteins [133]. Recently, the participation of Glo3p in virulence was proved in the nematode-trapping (NT) fungus, *Arthrobotrys oligospora*, which during the parasitic stage forms a complex three-dimensional network to trap nematodes, and where adhesion, penetration, and immobilization participate [134]. In *A. oligospora*, *Ao*GLO3 participates as a positive regulator of nematode-traps and therefore in the capture of nematodes. *Ao*GLO3 modulates positively the proteolytic activity of the serine protease PII as well [135], which is involved in the immobilization of nematodes and degradation of proteinaceous components of the nematode cuticle [136].

## 5. The Role of Small GTPases in Fungal-Plant Interactions

The role of small GTPases has been characterized in the fungus–plant relationship in both beneficial and pathogenic interactions. In fungi, the small GTPases most characterized have been proteins of the Ras and Rho families [137], which are involved in the formation of virulence structures [137,138,139], cell–cell fusion [140], and the production of ROS species [141], which are key biological functions in plant colonization by fungi.

### 5.1. Pathogenic Relationships

In *Ustilago maydis*, the causal agent of common smut of corn, RAC1 and Cdc42 are required for tumor formation on maize seedlings. When mixtures of the Δ*cdc42* and Δ*rac1* mutants are injected separately into maize plants, no tumor formation is observed; however, when mixtures of the mutants with the wild-type strain are injected, disease symptoms could be observed. Deletion of *cdc42* and *rac1* in a haploid strain did not result in tumor formation in maize. Therefore, the loss of virulence was a result of the fusion defect in the mutants [140].

In *B. cinerea* a foliar pathogenic fungus, BcRAS1 and BcRAC GTPases are involved in growth and virulence. BcRAC participates in the secretion of necrotic agents, essential for fungal virulence [105]. Moreover, Rho3 plays a fundamental role in appressoria formation, and virulence of *B. cinerea* on tomato plants [137]. Consistently, small GTPases Rho3, Erl, MoMsb2, and Ras2 are involved in the development of the appressorium and the virulence of the rice pathogen, *M. oryzae* [138,139].

*Colletotrichum gloeosporioides* (Anthracnose) is the causal agent of bitter rot in variety of crops, mainly perennials in tropical regions, including citrus, papaya, avocado, yam sweet pepper, and tomato among others. CgRac1 from *C. gloeosporioides* f. sp. aeschynomene regulates its virulence and appressoria development in the legume weed (*Aeschynomene virginica*) [142]. *Curvularia lunata* is a major pathogen causing leaf spot in maize. In this fungus, Clg2p, the homologue of Ras, regulates its virulence on corn inbred line Huangzao 4 [143].

Autophagy is involved in the infection and regulation of hyphal fusion in *Fusarium*, and several Rab GTPases have been considered to play an important role in autophagosome formation. In *F. graminearum*, FgAtg9 functions in delivering membranes for autophagosome formation and is essential for growth and pathogenicity. FgAtg9 interacts with FgRab7 to regulate FgAtg9 trafficking, which is essential for autophagy and pathogenicity. Furthermore, there is a conserved function of Atg9 in filamentous fungi because MoAtg9 from *M. oryzae* rescues the virulence of a Δ*Fgatg9* strain [144].

As discussed above, protein farnesylation regulates subcellular localization and interaction with other proteins. Farnesylation of the target protein is catalyzed by farnesyltransferase (FTase) composed of α-subunit Ram2 and β-subunit Ram1. In the rice blast fungus *M. oryzae*, Ram1 participates in hyphal growth, appressoria development, and virulence. Ras1 and Ras2 contain the C-terminal CAAX motifs and interact with Ram1 to be farnesilated, regulating their cellular localization in the plasma membrane of the appressorium [145].

Septins are a family of GTP-binding proteins conserved in all eukaryotic organisms. Septins are involved in cytokinesis, secretion, polarity determination, and reorganization of the cytoskeleton to determine cell shape. In *M. oryzae*, Sep3, Sep4, and Sep5 play a role in cell division and organization of the appressorium, whereas Cdc42 participates in septin ring formation. All these proteins regulate virulence in rice blast fungus. In conclusion, Cdc42 is required for septin ring assembly, which has repercussions in appressoria formation and virulence [146]. In support of these data, the small GTPase Tem1 from *S. cerevisae* is involved in cytokinesis and controls the septin dynamics to trigger ring contraction; hence, the septin signaling pathway could be regulated by small GTPases [147].

### 5.2. Mutualistic Relationships

The functions of small GTPases have also been studied in beneficial fungus–plant relationships, although to a lesser extent. *Trichoderma* spp. are filamentous fungi used as biostimulants and biocontrol agents since they are capable of promoting growth and protecting plants against pathogens. The product of the *tbrg*-1 gene from *T. virens* is involved in establishing a beneficial relationship with plants. The lack of *tbrg-1* affects the biocontrol activity of *T. virens* since the percentage of germination of tomato seeds treated with the Δ*tbrg-1* mutant and challenged with *R. solani* is lower than those treated with wt. Similarly, tomato seedlings treated with Δ*tbrg-1* are more susceptible to infection by fungal phytopathogen, suggesting that the lack of *tbrg-1* is a negative regulator of a potential pathogenic behavior of *T. virens* [15].

In the ectomycorrhiza *L. bicolor*, the *Ras1*-like homologue expression is dependent on the presence of its plant symbiotic partner. In this regard, free-living *L. bicolor* colonies do not show *LbRas* transcription. On the contrary, the presence of red pine roots enhances the transcription of *LbRas* [148]. Immunolocalization analysis shows that LbRas is mainly localized to the inner surface of the cell wall, presumably at the plasma membrane, which correlates with the presence of the “dual cysteine” residues in the palmitoylation motif on the LbRas HVR.

Fungi produce ROS during their interaction with plants. GTPases of the Rho family play important roles in the generation of ROS through Nox [149]. The Nox complex catalyzes the molecular conversion of oxygen to superoxide in an NADPH-dependent reaction [150]. In mammals, superoxide production depends on the association of the protein gp91phox with the small GTPase Rac1 and the heterotrimer composed of p67phox (Nox activator), p47phox (Nox organizer), and p40phox. NoxR is a fungal protein analogous to p40phox and p47phox and the fungal protein RacA is a homologue to Rac1. NoxR and RacA are essential components in maintaining symbiotic relationships with plants [150]. *Epichloe festucae* is a biotrophic filamentous fungus that forms symbiotic associations with perennial ryegrass (*Lolium perenne* cv. Yatzukase) [140,151]. NoxR interacts with Cdc24, which activates RacA. At the same time, RacA activates NoxA to maintain a symbiotic interaction with perennial ryegrass, by regulating cell–cell fusion, hyphal growth pattern, and host root colonization [140,152]. In conclusion, a control of hyphal growth regulated by small GTPases is required for symbiotic infection, and Cdc42 plays a more important role than RacA [140].

## 6. Conclusions

Since the discovery of small GTPases several decades ago, substantial advances have been made regarding their mechanism of regulation, their interaction with other proteins, and their function in fungi. In filamentous fungi, this group of proteins and their effectors control several signaling networks involved in growth and development and in response to internal and external stimuli as well as in their association with other organisms. Despite considerable advances, there are still major questions that require clarification and exploration in the different fungal models and outside of the models. An intriguing question is why some filamentous fungi bear a huge repertoire of small GTPases (many of them paralogous), which probably depends on the lifestyle they acquired through evolution. Another interesting question is whether those paralogues play unique or shared roles between them. The differences in regulation have been attributed to posttranslational regulation (i.e., lipidation), to differences in their promoters, and, importantly, to their downstream effectors. The signaling networks controlled by these proteins deserve attention since they regulate virulence in human, plant, and insect pathogens as well as in beneficial fungi to plants.

Analysis of the structure and function of small GTPases in virtue of approaches of omics, molecular biology, biological chemistry, and genetics will contribute to our understanding of their molecular mechanisms of regulation and their participation in several signal transduction pathways.

## Figures and Tables

**Figure 1 cells-10-01039-f001:**
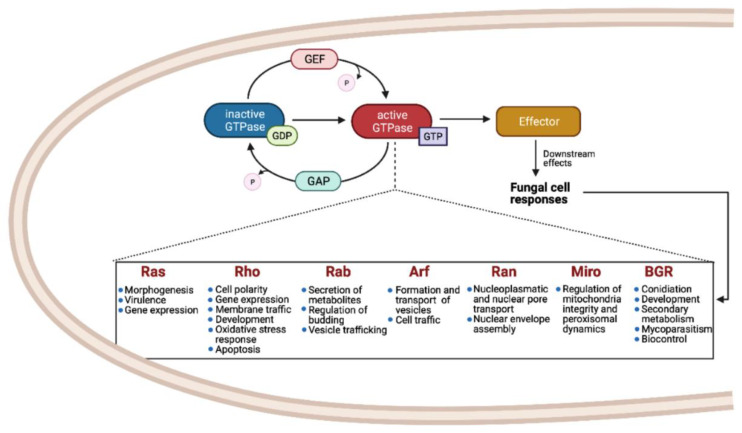
Small GTPases of the Ras superfamily regulate several processes in fungi. In fungal cells, GEFs and GAPs catalyze alternation between an active GTP-bound state and an inactive GDP-bound state. Active GTPase interacts with specific downstream effector, leading to different cell responses. The most representative functions of Ras GTPase families are listed in the box.

**Figure 2 cells-10-01039-f002:**
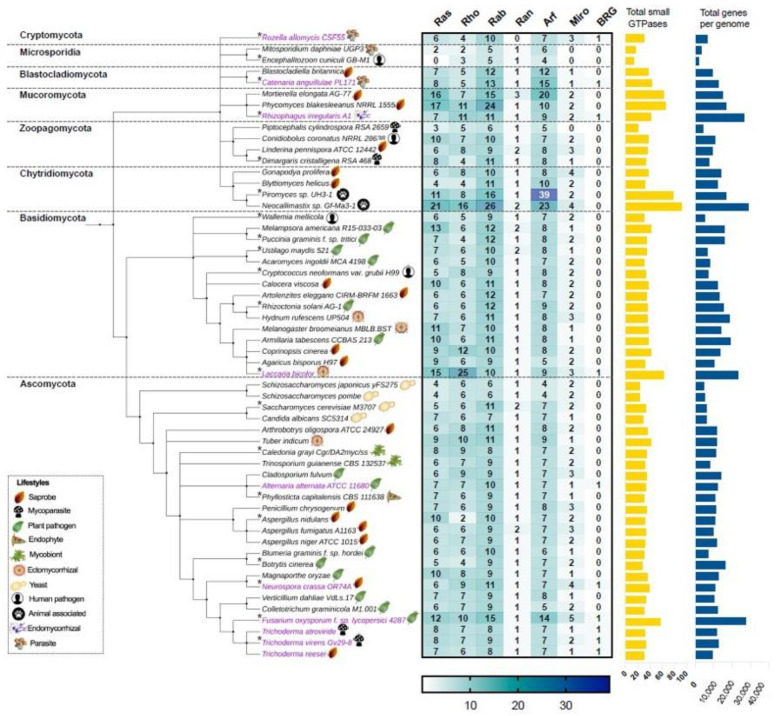
The Ras, Rho, Rab, Ran, Arf, Miro, and BRG families of the small Ras GTPases superfamily are present in the fungal kingdom. The taxonomic tree was generated using phyloT based on the NCBI taxonomy. Each phylum is indicated in its corresponding cluster. The numbers of members of each family are indicated in the heat-map table, which were obtained by searching in MycoCosm of JGI using PF00071, PF00025, and PF08477 keys as search criteria. The total number of small GTPases for each species are indicated in the yellow bars. The total number of genes of each genome were also obtained in MycoCosm and are indicated in blue bars. A representative lifestyle for each strain is indicated with icons. Those strains that contain putative BRGs in their genomes are indicated in magenta. The asterisks indicate the 20 species that were selected for further analysis.

**Figure 3 cells-10-01039-f003:**
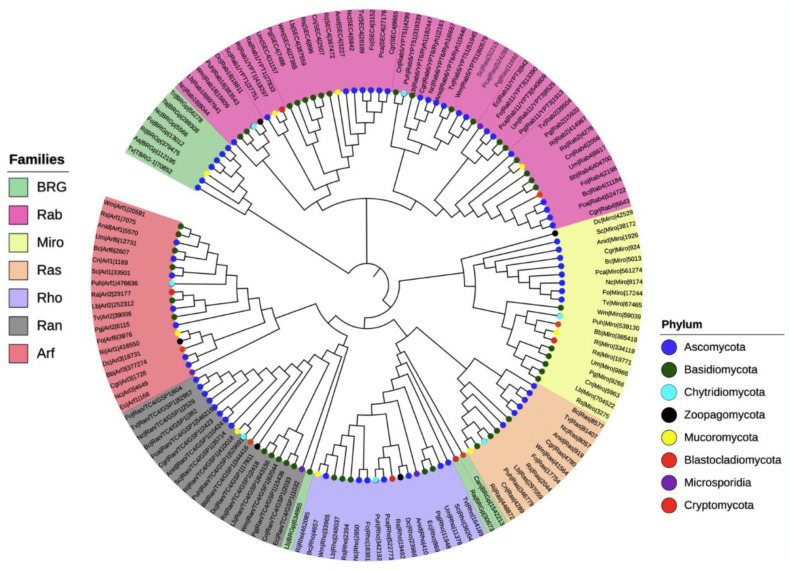
Phylogenetic tree of the seven families of the small Ras GTPases in fungi. The evolutionary history was inferred using the neighbor-joining method [27] using 144 proteins. The bootstrap test was used with 1500 replicates [28]. The evolutionary distances were computed using the Poisson correction method. Evolutionary analyses were conducted in MEGA7 [29]. Colored circles indicate each phylum. For full species names, please refer to Appendix A.

**Figure 4 cells-10-01039-f004:**
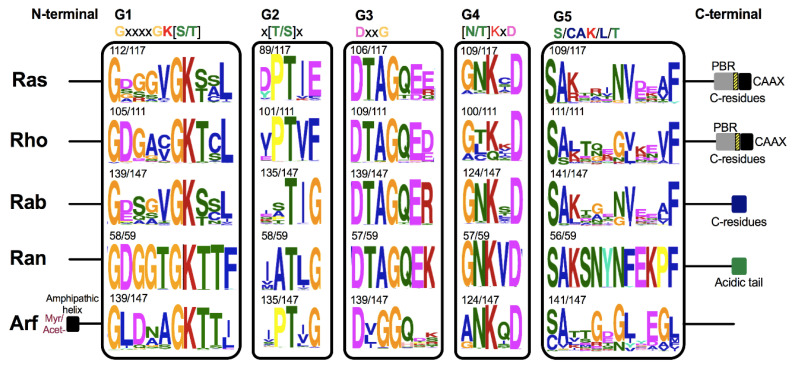
Schematic representation of the Ras, Rho, Rab, Ran, and Arf families’ motifs and their biochemical features for their post-translational modifications. The G1 to G5 motifs for each family are within the rectangles and each of the consensus sequences are indicated above. The total of motifs found among the proteins analyzed for each family is shown on top of each motif (number of motifs found/total proteins analyzed). The analysis was carried out on MEME-suite [35]. The amphipathic helix for Arf proteins is indicated at N-terminal as well as their possible myristoylation or acetylation sites. At the C-terminal, the PBR (gray rectangles), the C-residues for palmitoylation in the HVR (black and yellow rectangles with stripes), and the CAAX motif (black rectangles) for Ras and Rho families are indicated, as well as the C-residues motifs (blue rectangles) of Rab and the acidic tail (green rectangle) of Ran proteins are indicated.

**Figure 5 cells-10-01039-f005:**
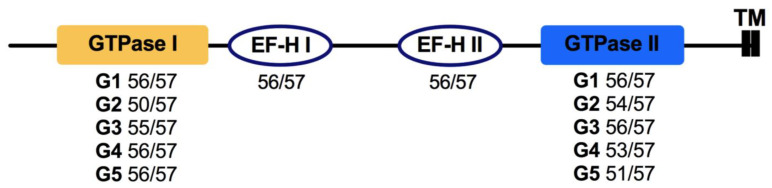
Schematic representation of conserved proteins domains of Miro family. The GTPase I and II domains are indicated in the rectangles, whereas the EF-H I and II domains are indicated in the ovals. Below each motif, the number of each motif found among the 57 proteins analyzed is shown. The transmembrane domain (TM) at the C-terminal present in Miro proteins is also indicated. The distribution of each motif in sequences was drawn according to MEME results.

**Figure 6 cells-10-01039-f006:**
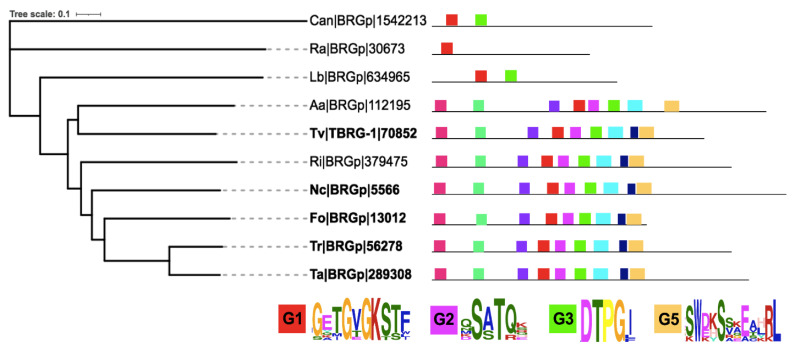
The BRG family is mainly present in Sordariomycetes. The evolutionary history was inferred using the neighbor-joining method [27] using the 10 putative BRG proteins from Cryptomycota, Blastocladiomycota, Mucoromycota, Basidiomycota, and Ascomycota phyla. The bootstrap test was applied using 1500 replicates [28]. The tree is drawn to scale, with branch lengths in the same units as those of the evolutionary distances used to infer the phylogenetic tree. The evolutionary distances were computed using the Poisson correction method. Evolutionary analyses were conducted in MEGA7 [29]. The motifs’ locations according to MEME results are indicated to the right of the tree, where each colored box represents a motif or conserved region. The consensus sequences obtained from the MEME-suite [35] of each motif found are shown together with each corresponding colored box. The height of each amino acid corresponds to its frequency of occurrence in the alignment. Those strains belonging to Sordariomycetes are indicated with a bold letter.

**Figure 7 cells-10-01039-f007:**
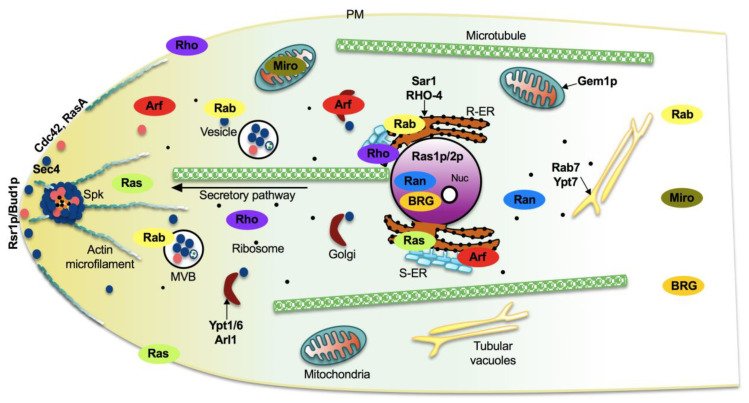
Subcellular localization of Ras, Rho, Rab, Ran, Arf, Miro, and BRG GTPases. The localization of each GTPase depends mainly on its activity. Ras (green ovals) and Rho (purple ovals) proteins are primarily found at ER, cytoplasm, and PM; Rab (light yellow ovals) at ER, vesicles, MVB, or endosomes; Ran (blue ovals) at nucleus and cytoplasm; Arf (red ovals) at ER, Golgi, and cytoplasm; Miro (olive green ovals) at mito-chondria, and BRG (dark yellow ovals) are predicted to be found mainly in the nucleus. Rab, Miro, and BRG could be present at the cytoplasm as well but to a lesser extent. Some examples of specific small Ras GTPases are shown (i.e., Ras, Rsr1p/Bud1p, RasA, Ras1p/2p; Rho, Cdc42, RHO-4; Rab, Sec4, Rab7, Ypt1/6, Ypt7; Arf, Sar1, Arl1; Miro, Gem1p). This figure was drawn according to the literature discussed here as well as by predicting their subcellular localization using WoLF PSORT [57] and COMPARTMENTS [58] programs. The names of each organelle are indicated. Spk: Spitzenkörper; S-ER, R-ER: smooth and rough endoplasmic reticulum; PM: plasma membrane; Nuc: nucleus.

## Data Availability

All data are included in the manuscript.

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
