# Peer review of "The Small GTPases in Fungal Signaling Conservation and Function"

_cells, 2021, doi:10.3390/cells10051039_

Round 1

Reviewer 1 Report

The manuscript at hand gives an overview on the phylogeny and function of small GTPases in fungi. The first two chapters provide an extensive bioinformatic analysis of selected fungal genomes from databases with regard to the occurence and domain structure of small GTPases. Chapter 3 then summarizes some of the biological functions in detail, while ignoring others. In chapter 4 the authors then shortly address the role of the GTPases both in pathogenic and mutualistic interactions of the host fungi with plants.

In general, a review focussing on small GTPases in fungi is suitable for the journal and may attract a sufficiently broad readership. It drew my attention that the authors seem to have published very few original work on the subject, though. Nevertheless, several points should be addressed prior to publication, as outlined in the following.

  1. So English is fairly good, there are a large number of phrasing errors, so that I strongly suggest to submit the manuscript to an editing service.
  2. The authors tend to insert a lot of unnecessary terms, like “In this way, ...” (line 69), ”In this sense ...” (line 72), “establish cross-talk” (line 55, which could simply read “cross-talk to each other”) etc.; this should be carefully checked and eliminated throughout the manuscript.
  3. The statement of being the first to review the subject “in many species of fungi” (line 58) should also be eliminated. It does not help the reader and actually seems a little overstated, given that hundreds of reviews are available on GTPases, though they may have a different focus.
  4. What is meant by the sentence in line 210 “as well as the coupling of residues immediately upstream, such as C modified by ...”. Such as a cysteine to which palmitate can be attached? – Where is this cysteine residue in Figure 3? -Inclusion of this and the PBR region (that is how the polybasic region is usually called) in Figure 3 would help to clarify a lot of this discussion.
  5. This is also true for the preceeding sequences. I suppose that HRV (line 218) should be HVR for hypervariable region?
  6. line 226 should read “at their C-terminal end”. Are the remaining 35 proteins really devoid of the CAAX sequence?
  7. line 243: Rac1 is distinguished here from the other Rho family members. However, it is assumed that at least Rho5 from S. cerevisiae is a Rac homolog. Authors should refer to these papers.
  8. Figure 10 should be revised. Information on the functions of the GTPases is not readable at the original size and enlarging the PDF file results in poor letter quality. There is a lot of empty space in this figure, so that the table can be enlarged without any problem.
  9. Again, in the functions listed in Figure 10 the role of Rho5 in oxidative stress response and apoptosis established for the yeast protein (Singh et al. 2008, PNAS 106; Schmitz et al. 2015, Molec. Microbiol. 96) should be included.
  10. line 627: the term “the filamentous fungal human pathogens” is misleading. Both species are known to grow in a yeast form and are at best dimorphic. They are not filamentous per se, although they may form filaments in the human body.
  11. line 652: “which potentially codes for a GAP inhibitor of RasA”, should read “which presumably encodes a GAP for RasA”; it could be added “deactivating the GTPase” if necessary. Inhibition is usually an allosteric process and different from deactivation!
  12. line 721: The paragraph on gossypii (not the right scientific name, anyway!) completely ignores the knowledge on two Rho1 GTPases published.
  13. From chapter 3.3. until the beginning of chapter 4 all species names should be written in italics.

More generally, the review would gain much from considerable condensation. Thus, in the first half the authors may try to avoid listing numbers that are already evident in the figures and instead reduce them to a few conclusive sentences. It is also questionable, if textbook knowledge on phylogenetic relationships amongst fungi needs to be discussed in the context of this review. Chapters 3 and 4 also list anecdotal evidence from different species, rather than giving the impression of an ordered and inherent story. With these hints, I would suggest that the authors try to condense the information and reduce the number of pages by at least 15%.

Author Response

Dear Professor Bor Luen Tang,

Please, find attached the answers to the questions and comments of the reviewers of our manuscript entitled: “The small GTPases in fungal signaling conservation, function, and regulation” by Mitzuko Dautt-Castro, Montserrat Rosendo-Vargas, and myself, which was previously submitted to Cells Journal as Manuscript ID: cells-1163961.

Reviewer 1

The manuscript at hand gives an overview on the phylogeny and function of small GTPases in fungi. The first two chapters provide an extensive bioinformatic analysis of selected fungal genomes from databases with regard to the occurence and domain structure of small GTPases. Chapter 3 then summarizes some of the biological functions in detail, while ignoring others. In chapter 4 the authors then shortly address the role of the GTPases both in pathogenic and mutualistic interactions of the host fungi with plants.

In general, a review focussing on small GTPases in fungi is suitable for the journal and may attract a sufficiently broad readership. It drew my attention that the authors seem to have published very few original work on the subject, though. Nevertheless, several points should be addressed prior to publication, as outlined in the following.

  1. So English is fairly good, there are a large number of phrasing errors, so that I strongly suggest to submit the manuscript to an editing service.

R = We let the editorial office about our intention to send our manuscript for English editing by a native speaker, however, they suggest us don't do that, since accepted manuscript are submitted for language revision by cells editorial office.

  1. The authors tend to insert a lot of unnecessary terms, like “In this way, ...” (line 69), ”In this sense ...” (line 72), “establish cross-talk” (line 55, which could simply read “cross-talk to each other”) etc.; this should be carefully checked and eliminated throughout the manuscript.

R = We already eliminated most of the unnecessary terms as suggested by the reviewer, however, we conserved some that we consider are necessary to connect some sentences.

  1. The statement of being the first to review the subject “in many species of fungi” (line 58) should also be eliminated. It does not help the reader and actually seems a little overstated, given that hundreds of reviews are available on GTPases, though they may have a different focus.

R = Thank you for the advice, we already eliminate such sentence as suggested.

  1. What is meant by the sentence in line 210 “as well as the coupling of residues immediately upstream, such as C modified by ...”. Such as a cysteine to which palmitate can be attached? – Where is this cysteine residue in Figure 3? -Inclusion of this and the PBR region (that is how the polybasic region is usually called) in Figure 3 would help to clarify a lot of this discussion.

R = The statement of the reviewer is right; the cysteine is the amino acid where the palmitate is added. Usually, this consists of a two cysteines motif. We added this information to the manuscript to clarify it. Also, we indicate that the polybasic region is also called PBR, and we added the localization of the PBR and C-residues features in Figure 3. As you can notice, we also modified the Figure 3 as suggested for another reviewer.

  1. This is also true for the preceeding sequences. I suppose that HRV (line 218) should be HVR for hypervariable region?

R = The reviewer is right; the error has been corrected.

  1. line 226 should read “at their C-terminal end”. Are the remaining 35 proteins really devoid of the CAAX sequence?

R = We appreciate the reviewer observation. We already changed such sentence as suggested by the reviewer. Also, we analyzed again the Ras proteins and observed that 83 out of 117 proteins contain the CAAX sequence. The remaining 34 proteins indeed lack this motif. Curiously, we figure out that 16 of these 34 proteins lacking the CAAX motifs contain a C residue at their C-terminal end, which is a characteristic of the Rho family. This could indicates that their annotation in JGI is not accurate. This fact correlates well with our phylogenetic analysis, where three proteins annotated in JGI as Ras were grouped with the Rab family (gray labels, Figure 2), and these three proteins lack of CAAX motif.

  1. line 243: Rac1 is distinguished here from the other Rho family members. However, it is assumed that at least Rho5 from S. cerevisiae is a Rac homolog. Authors should refer to these papers.

R = We appreciate the reviewer observation. The homology between Rho5 and Rac has been clarify in the manuscript, according to Elias and Klimes (2012), who performed a different phylogeny analysis of Rho family.

  1. Figure 10 should be revised. Information on the functions of the GTPases is not readable at the original size and enlarging the PDF file results in poor letter quality. There is a lot of empty space in this figure, so that the table can be enlarged without any problem.

R = Thank you for the comment. We already tried to improve the quality of figure 10 and enlarge the table of small GTPases functions.

  1. Again, in the functions listed in Figure 10 the role of Rho5 in oxidative stress response and apoptosis established for the yeast protein (Singh et al. 2008, PNAS 106; Schmitz et al. 2015, Molec. Microbiol. 96) should be included.

R = Done

  1. line 627: the term “the filamentous fungal human pathogens” is misleading. Both species are known to grow in a yeast form and are at best dimorphic. They are not filamentous per se, although they may form filaments in the human body.

R = we already clarify such point. Thank you for the comment.

  1. line 652: “which potentially codes for a GAP inhibitor of RasA”, should read “which presumably encodes a GAP for RasA”; it could be added “deactivating the GTPase” if necessary. Inhibition is usually an allosteric process and different from deactivation!

R = Thank you for the advice, the change was made.

  1. line 721: The paragraph on gossypii (not the right scientific name, anyway!) completely ignores the knowledge on two Rho1 GTPases published.

R = We already the Scientific name for gossypii and added the missed information about Rho1. Thank you for your comment.

  1. From chapter 3.3. until the beginning of chapter 4 all species names should be written in italics.

R = We’re sorry about our careless, the species names in chapter 3 were already written in italics as suggested by the reviewer.

More generally, the review would gain much from considerable condensation. Thus, in the first half the authors may try to avoid listing numbers that are already evident in the figures and instead reduce them to a few conclusive sentences.

R = As suggested by the reviewer, we condensed bioinformatic analysis results by removing most of the numbers that appear in Figure 3. Also, by merging Figures 3 to 7 in only one allowed us to reduce some sentences related to these results. Moreover, we removed some information that could be repetitive or related with other sections. Thank you for the advice.

It is also questionable, if textbook knowledge on phylogenetic relationships amongst fungi needs to be discussed in the context of this review.

R = We thank for the suggestion. We included a textbook shown below, however we did not find additional useful information to discuss here. We think the relevant publications are already included.

Eliás, M. and Klimes, V. Rho GTPases: Deciphering the Evolutionary History of a Complex Protein Family. In Rho GTPases. Springer, New York. 2012, 827, 13-24, doi: 10.1007/978-1-61779-442-1_2 

Chapters 3 and 4 also list anecdotal evidence from different species, rather than giving the impression of an ordered and inherent story. With these hints, I would suggest that the authors try to condense the information and reduce the number of pages by at least 15%.

R = Thank you for the advice, we already reduced the number of pages as suggested.

Reviewer 2 Report

Casas-Flores and colleagues have complied different aspects of the GTPases of the RAS superfamily in organisms from different kingdoms, including RAS classification by using phylogenetics and sequences analysis of 56 different genomes belonging to different phyla.

Overall, a huge amount of information is put together, which was for sure a lot of work. However, the manuscript is very detailed and too comprehensive, and therefore often not substantial. For example, most of the statistics and the phenotype descriptions in different phyla are redundant, and could be summarized in one section without repeating it again and again.

What still stands out above all is the inconsistencies, especially the gene or protein names (Upper case or lower case, italics or not).

In addition to the strong shortening of the manuscript and merging Figures 3-7 into one, other figures, like subcellular localizations of the respective proteins and a table summarizing the protein functions in the different fungi families would considerably improve readability.

GTPase regulations have come up short. So, the title could be read “The small GTPases in fungal signaling conservation and function”.

A concluding remark section with future perspectives would be crucial. An interesting issue, for example, could be link between the abundance of number of members of RAS families and the life-cycle of the specific fungal species.

References:

A very recent fungi-related article nicely fits to this article:

Membrane traffic related to endosome dynamics and protein secretion in filamentous fungi. Higuchi Y. Biosci Biotechnol Biochem. 2021 Jan 11:zbab004.

With a total number of 155 references, there is definitely space for classical review articles that are quite central to this article, such as:

The GTPase superfamily: a conserved switch for diverse cell functions. Bourne HR, Sanders DA, McCormick F. Nature. 1990 Nov 8;348(6297):125-32.

The GTPase superfamily: conserved structure and molecular mechanism. Bourne HR, Sanders DA, McCormick F. Nature. 1991 Jan 10;349(6305):117-27.

The Ras superfamily of GTPases.  Macara IG, Lounsbury KM, Richards SA, McKiernan C, Bar-Sagi D.FASEB J. 1996 Apr;10(5):625-30.

Structure-function relationships of the G domain, a canonical switch motif. Wittinghofer A, Vetter IR. Annu Rev Biochem. 2011;80:943-71.

Author Response

Dear Professor Bor Luen Tang,

Please, find attached the answers to the questions and comments of the reviewers of our manuscript entitled: “The small GTPases in fungal signaling conservation, function, and regulation” by Mitzuko Dautt-Castro, Montserrat Rosendo-Vargas, and myself, which was previously submitted to Cells Journal as Manuscript ID: cells-1163961.

Reviewer 2

Casas-Flores and colleagues have complied different aspects of the GTPases of the RAS superfamily in organisms from different kingdoms, including RAS classification by using phylogenetics and sequences analysis of 56 different genomes belonging to different phyla.

  1. Overall, a huge amount of information is put together, which was for sure a lot of work. However, the manuscript is very detailed and too comprehensive, and therefore often not substantial. For example, most of the statistics and the phenotype descriptions in different phyla are redundant, and could be summarized in one section without repeating it again and again.

R = We appreciate the suggestion, which undoubtedly helped us to improve our manuscript. In concordance with your suggestion, we delete the redundant information related to phenotypes and phyla. We condensed the information by mentioning that only in section 2 and was deleted from the discussion on bioinformatic analysis. Also, we remove from section 3 some information that could be repetitive compared with other topics.

  1. What still stands out above all is the inconsistencies, especially the gene or protein names (Upper case or lower case, italicsor not).
    R = Thank you for the observation. Genes were written in lowercase and italics, whereas proteins were written in uppercase and plane letters. However, we consider important that some genes should remain written as the original articles to avoid confusion with the readers, since the authors handled them that way. For instance, RAS2 from reference (Bluhm et al. 2007); Mras1, Mras2, and Mras3 from reference (Wendland and Philippsen, 2001); RAS1 from reference (Park et al. 2006); Ras 1 and Ras 2 from references (Xie et al. 2013; Knabe et al. 2013); TrRas1 and TrRas2 from reference (Zhang et al. 2012) and FgMon1 from reference (Li et al. 2015).

  1. In addition to the strong shortening of the manuscript and merging Figures 3-7 into one, other figures, like subcellular localizations of the respective proteins and a table summarizing the protein functions in the different fungi families would considerably improve readability.

R = Figures 3-7 were merged into one which helped us to improve our manuscript. We hope now it is easier to read. Moreover, we included the figure 6 related to subcellular localization as suggested. Thank you for the advice.

About the suggested table to summarize the functions of small Ras GTPases, we believe that this could be redundant, because such information is listed in Figure 7.

  1. GTPase regulations have come up short. So, the title could be read “The small GTPases in fungal signaling conservation and function”.

R = The title was changed as suggested. We appreciate the advice.

  1. A concluding remark section with future perspectives would be crucial. An interesting issue, for example, could be link between the abundance of number of members of RAS families and the life-cycle of the specific fungal species.

R = We already add a concluding remarks section.

  1. A very recent fungi-related article nicely fits to this article:

Membrane traffic related to endosome dynamics and protein secretion in filamentous fungi. Higuchi Y. Biosci Biotechnol Biochem. 2021 Jan 11:zbab004.

With a total number of 155 references, there is definitely space for classical review articles that are quite central to this article, such as:

The GTPase superfamily: a conserved switch for diverse cell functions. Bourne HR, Sanders DA, McCormick F. Nature. 1990 Nov 8;348(6297):125-32.

The GTPase superfamily: conserved structure and molecular mechanism. Bourne HR, Sanders DA, McCormick F. Nature. 1991 Jan 10;349(6305):117-27.

The Ras superfamily of GTPases.  Macara IG, Lounsbury KM, Richards SA, McKiernan C, Bar-Sagi D.FASEB J. 1996 Apr;10(5):625-30.

Structure-function relationships of the G domain, a canonical switch motif. Wittinghofer A, Vetter IR. Annu Rev Biochem. 2011;80:943-71.

R = We are grateful for suggesting us to include some important references, which are cited now in the manuscript.

Finally, we would like to take this opportunity to thank the referees for their questions and comments that have allowed us to improve our manuscript. We hope you will find our work suitable for publication in Cells journal.

We are looking forward to hearing from you.

Sincerely yours,

Dr. J. Sergio Casas Flores

Principal Investigator

División de Biología Molecular

IPICYT

Reviewer 3 Report

In the manuscript titled “The small GTPases in fungal signaling conservation, function, and regulation”, authors Dautt-Castro et al., have detailed an overview of different small GTPases important in fungi. The authors write about the genomic, phylogenetic, and sequence analysis of each protein family. The article is well researched and lists all the members of the GTPases family that have been reported in fungi. The authors provide a review of the literature explaining the role played by various GTPases in pathogenic or mutualistic relationships. This review is a comprehensive and exhaustive summary of the fungal GTPases and their function.

Author Response

Dear Professor Bor Luen Tang,

Please, find attached the answers to the questions and comments of the reviewers of our manuscript entitled: “The small GTPases in fungal signaling conservation, function, and regulation” by Mitzuko Dautt-Castro, Montserrat Rosendo-Vargas, and myself, which was previously submitted to Cells Journal as Manuscript ID: cells-1163961.

Reviewer 3

In the manuscript titled “The small GTPases in fungal signaling conservation, function, and regulation”, authors Dautt-Castro et al., have detailed an overview of different small GTPases important in fungi. The authors write about the genomic, phylogenetic, and sequence analysis of each protein family. The article is well researched and lists all the members of the GTPases family that have been reported in fungi. The authors provide a review of the literature explaining the role played by various GTPases in pathogenic or mutualistic relationships. This review is a comprehensive and exhaustive summary of the fungal GTPases and their function.

Dear reviewer,

Thank you for taking the time for reviewing our manuscript.

Finally, we would like to take this opportunity to thank the referees for their questions and comments that have allowed us to improve our manuscript. We hope you will find our work suitable for publication in Cells journal.

We are looking forward to hearing from you.

Sincerely yours,

Dr. J. Sergio Casas Flores

Principal Investigator

División de Biología Molecular

IPICYT

Round 2

Reviewer 1 Report

Provided that English is considerably corrected in the editing process as claimed by the authors, I have no further objections to the revised version of the manuscript.

Author Response

Dear reviewer,

Please find below our responses to your suggestions. We really appreciate your valuable revision.

Line 29: the Overview starts with Figure 7. Maybe it could be named Figure 1.

R= We appreciate your comment. Figure 7 was placed as Figure 1 and consequently the different figures were ordered based on such change.

Line 36: "switches I and II regions" should read "switch I and II regions".

R= The sentence was changes as suggested.

Line 237: Please note that not "hypervariable region (HVR)" contain "a 
poly-basic region (PBR)". So, it is better to name this region HVR instead of PBR (throughout the manuscript).

R= We apologize for the confusion, the sentence “…and bears an hypervariable region (HVR) which codes a poly basic region (PBR)”, the last part: “which codes a poly basic region (PBR)” was removed.

Furthermore, we added some information to clarify the information regarding the hypervariable region (HVR) versus the polybasic region (PBR). As mentioned in section 3.1, when we described the Figure 3 (now Figure 4) throughout the manuscript, we specifically refer to polybasic region (PBR), basically because in our analysis we search those lysines or arginines residues on the Ras and Rho sequences at the C-terminal. For this reason, we prefer to be specific and designate it as PBR. Also, we distinguish in a specific way those C-residues related to palmitoylation that are presents in the HVR.

Besides the references related to this topic already cited in the manuscript, we included below two more works, which we think are relevant to clarify this issue.

Hancock, J. F.; Magee, A. I.; Childs, J. E.; Marshall, C. J. All ras proteins are polyisoprenylated but only some are palmitoylated. Cell. 198957(7), 1167-1177, doi: 10.1016/0092-8674(89)90054-8

Hancock, J. F.; Paterson, H.; Marshall, C. J. A polybasic domain or palmitoylation is required in addition to the CAAX motif to localize p21ras to the plasma membrane. Cell. 199063(1), 133-139, doi: 10.1016/0092-8674(90)90294-O

Line 453: Figure 5 might be better organized in order to increase its 
overall size.

R= Thank you for the advice. The figure was improved, and its size was increased as suggested. We hope you’ll find now suitable for the manuscript.

Line 483: "...depends mainly on their activity" should read "...depends 
mainly on its activity.

R= Thanks for the suggestion, the sentence was written as suggested.

Line 542: "Ras GTPases regulate several processes in fungi" may be changed to "Small GTPases of the RAS superfamily regulate several processes in fungi".

R= We agree with you, the sentence was corrected as suggested.

Line 582: "modulator of RasA" should read "regulator of RasA".

R= The sentence was changed as suggested